# Upper Atmosphere Responses to the 2022 Hunga Tonga-Hunga Ha'apai Volcanic Eruption via Acoustic-Gravity Waves and Air-Sea Interaction

Qinzeng Li[1,5], Jiyao Xu[1,2]*, Aditya Riadi Gusman[3], Hanli Liu[4], Wei Yuan[1,5], Weijun Liu[1,5], Yajun Zhu[1,5], and Xiao Liu[6]

1. State Key Laboratory of Space Weather, National Space Science Center, Chinese Academy of Sciences, Beijing, 100190, China

2. School of Astronomy and Space Science, University of Chinese Academy of Science, Beijing, 100049, China

3. GNS Science, Lower Hutt, New Zealand

4. High Altitude Observatory, National Center for Atmospheric Research, Boulder, Colorado, USA

5. Hainan National Field Science Observation and Research Observatory for Space Weather, National Space Science Center, Chinese Academy of Sciences, Beijing, 100190, China

6. School of Mathematics and Information Science, Henan Normal University, Xinxiang, China

Corresponding author: Jiyao Xu (jyxu@swl.ac.cn)

# Abstract

Multi-group of strong atmospheric waves (wave packets #1-#5) over China associated with the 2022 Hunga Tonga–Hunga Ha'apai (HTHH) volcano eruptions were observed in the mesopause region using a ground-based airglow imager network. The horizontal phase speed of wave packet #1 and #2 is approximately 309 m/s and 236 m/s respectively, which is consistent with Lamb wave L0 mode and L1 mode from theoretical prediction. The amplitude of the lamb wave L1 mode is larger than that of L0 mode. The wave fronts of Lamb wave L0 and L1 below the lower thermosphere are vertical, while the wave fronts of L0 mode tilt forward above exhibiting internal wave characteristics, which show good agreement with the theoretical results. Two types of tsunamis were simulated, one type of tsunami is induced by the atmospheric pressure wave (TIAPW) and the other type tsunami is directly induced by the Tonga volcano eruption (TITVE). From backward ray tracing analysis, the TIAPW and TITVE were likely the sources of the wave packet #3 and wave packets #4-5, respectively. The scale of tsunamis near the coast is very consistent with the atmospheric AGWs observed by the airglow network. The AGWs triggered by TITVE propagate nearly 3000 km inland with the support of duct. The atmospheric pressure wave can directly affect the upper atmosphere, and can also be coupled with the upper atmosphere through the indirect way of generating tsunami and subsequently tsunami generating AGWs, which will provide a new understanding of the coupling between ocean and atmosphere.

## 1. Introduction

Hunga Tonga–Hunga Ha'apai (HTHH) volcano, which erupted at 04:14:45 UT on January 15, 2022, produced the largest volcanic eruption in terms of energy release of a single event since the Krakatoa volcanic eruption (Symons, 1888) in 1883. This volcanic eruption triggered broad spectrum atmospheric disturbances (Adam, 2022; Duncombe, 2022; Wright et al., 2022), including Lamb waves (Zhang et al., 2022), acoustic waves, gravity waves (GWs) (Liu et al., 2022), and shock waves (Astafyeva et al., 2022). In addition, the travelling ionospheric disturbances (TIDs) caused by this volcanic eruption have also been reported (Themens et al., 2022; Lin et al., 2022).

Lamb waves are external wave propagating along Earth's surface at the speed of sound (Beer, 1974). They are non-dispersive or nearly non-dispersive (Francis, 1973) and can propagate horizontally over long distances. Lamb wave mainly occupies the troposphere, and its perturbation pressure decays exponentially with height (Yeh and Liu, 1974). The Lamb waves excited by the Tonga volcano eruptions went around the Earth several times (Amores et al., 2022; Duncombe, 2022). Sepúlveda et al. (2023) found that the wind field strongly affects the morphology and propagation of Lamb wave. Liu et al. (2023) reproduced the Lamb wave L0 and L1 modes consistently with theoretical predictions (Francis, 1973) using high-resolution Whole Atmosphere Community Climate Model with thermosphere/ionosphere extension (WACCM-X). Li et al. (2023) identified Lamb wave L1 mode using phase-leveling amplitude technology based on global navigation satellite system (GNSS)-total electron content (TEC). Poblet et al. (2023) reported that the strong perturbations in the meteor radar horizontal wind field over South

America is caused by lamb wave L1 mode associated with the 2022 HTHH volcano eruption.

Acoustic-gravity waves (AGWs) are mechanical waves in compressible fluids in a gravity field (Gossard and Hooke, 1975). If the frequencies are much larger than the buoyancy frequency, AGWs tend towards acoustic wave mode, and when the frequency is much smaller than the buoyancy frequency, the fluid can be considered incompressible, and the AGWs tend towards internal GWs mode. The term "acoustic-gravity waves" is usually used when restoring forces due to both gravity and compressibility are important. AGWs are known to play a significant role in the coupling between the atmosphere/ionosphere and the ocean (Press and Harkrider, 1962; Harkrider and Press, 1967; Donn and Balachandran, 1981; Azeem et al., 2017). Atmospheric pressure waves are mechanical waves that are related to the density of the atmosphere. Compression and expansion are the high-pressure and low-pressure regions of motion in a medium.

The 2022 HTHH volcano eruption triggered tsunamis that affected the whole world (Carvajal et al., 2022; Ghent et al., 2022). Conventional tsunamis are typically generated by localized sea surface displacements caused by sources such as earthquakes and volcanoes, similar to the tsunamis directly induced by the 2022 Tonga volcano eruption (TITVE). Another tsunami is induced by the atmospheric pressure wave (TIAPW) (Kubota et al., 2022; Gusman et al., 2022). Tsunami can generate upward propagating AGWs through water-air interface and propagate to the thermosphere/ionosphere (Hines, 1972; Peltier and Hines, 1976; Hickey et al., 2009, 2010; Occhippinti et al., 2013; Vadas et al., 2015; Laughman et al., 2016; Nishikawa et al., 2023; Pradipta et al., 2023). Using the red line

airglow imager, Makela et al. (2011) detected airglow disturbance in Hawaii that arrived

1hr earlier of the tsunami generated by the 11 March 2011 Tohoku earthquake. Also using

the redline airglow, Smith et al. (2015) observed tsunami and GW almost simultaneously in

Chile. Inchin et al. (2020) used a three dimensional (3D) numerical model to simulate the

atmospheric AGWs generated by tsunami. They found that bathymetry variations

significantly affected the tsunamis and the AGWs excited by tsunamis, leading to their

nonlinear evolution process. More recently, Inchin et al. (2022) performed the numerical

simulations of mesopause airglow radiation fluctuations induced by tsunami-generated

AGWs, and found that large-scale tsunamis can cause detectable and quantitative

disturbances of mesopause airglow through AGWs.

As far as we know, the research on the impact of tsunamis induced atmospheric

AGWs on the atmosphere and ionosphere shown above is all caused by conventional

tsunami. There are only two rare studies on the ground-based airglow observations of

AGWs caused by this conventional tsunami, and both are limited to red line observations

(Makela et al., 2011; Smith et al., 2015). However, the observation of tsunami induced

AGWs in the mesopause region observed by ground-based airglow imaging has never been

reported. In this study, we first reported the propagation characteristics of the AGWs

generated by the tsunamis triggered by the 2022 HTHH volcano eruptions in the

mesopause region using the ground-based airglow imager observation network. We then

focus on the coupling process of atmospheric pressure waves triggering tsunamis, and then

tsunamis generating atmospheric AGWs through air-water-air-coupling process in the

far-field area of the 2022 HTHH volcano eruption.

## 2. Data and Methods

### 2.1 Multi layer airglow imager network

A multi-layer airglow observation network (Xu et al., 2021) was built to study atmospheric disturbances excited by severe weather events, such as thunderstorms (Xu et al., 2015), typhoons (Li et al., 2022) and volcanic activities. Figure 1 shows the distribution of the multi-layer airglow observation network station. The multi-layer airglow observation network mainly includes the OH airglow network, which has been used to observe the airglow layer at the height of 87 km; the OI airglow network has been used to observe the airglow layer at the height of 250 km. In addition, there were 557 nm airglow and Na airglow imagers installed at some stations, such as Xinglong Station (40.4°N, 117.6°E), Lhasa (29.7°N, 91.0°E). The airglow network can provide observation with high temporal and spatial resolution. The temporal resolution is 1 min and the spatial resolution is 1 km. The time resolution of OH airglow imager is 1 minute, while the resolution of OI 557 nm and OI 630 nm airglow imager is 3 minutes, respectively. The spatial resolution of the airglow imager at the airglow layer is not uniform. The resolutions of OH, OI 557 nm, and OI 630 nm airglow in the zenith direction are 0.27 km, 0.29 km, and 0.77 km, respectively, while in the zenith angle of 60°, the resolutions are 1.01 km (OH), 1.11 km (OI 557 nm), and 2.65 km (OI 630 nm), respectively.

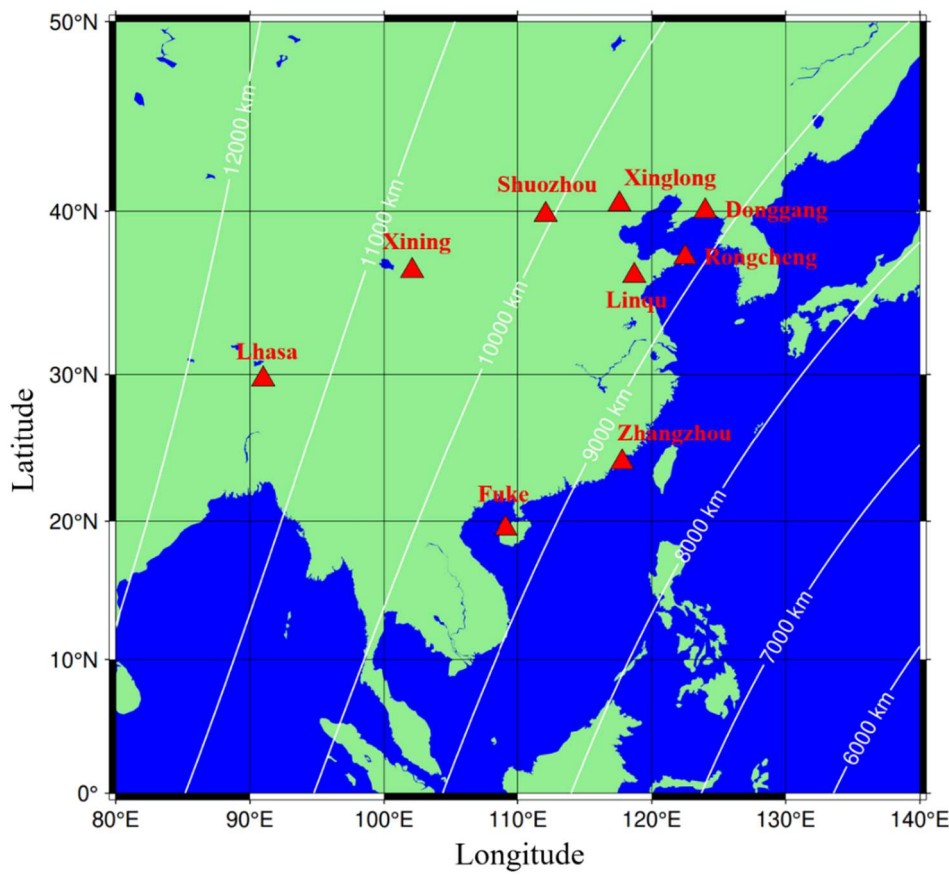

**Figure 1** The distribution of airglow network stations, along with the large circular centered on the Tonga volcano and its radius length, is also marked in the figure.

## 2.2 Spectral analysis of atmospheric wave parameters

The airglow image was calibrated with the help of standard star map (Garcia et al., 1997) and projected into geospatial space. The background radiation is removed by time differential (TD) method (Swenson and Mende, 1994), to highlight atmospheric fluctuations. The atmospheric wave parameters (horizontal wavelength $\lambda_h$ , observed horizontal phase speed $c$ , and the relative intensity perturbation $I'/I$ ) are extracted from spectral analysis method. Figure 2c presents the two-dimensional cross spectrum obtained from Fig. 2a and 2b. Zonal ($k_x$) and meridional ($k_y$) wave numbers are determined from the peak position of the spectra. The horizontal wavelengths $\lambda_h$ are obtained from the expression of $\lambda_h = 2\pi / \sqrt{k_x^2 + k_y^2}$ . The observed speeds $c$ are calculated from the phase ($\varphi$ )

(Fig. 2d) at the maximum peak of the cross spectrum as $c = \dfrac{\varphi}{2\pi}\dfrac{\lambda_h}{\Delta t}$, where $\Delta t$ is the time
interval between the two TD images. The amplitudes of intensity perturbations were
calculated by integrating the power surrounding the central peaks of the power spectrum.
To eliminate noise, the energy of the wave spectrum should be greater than 10% of the total
spectrum (Tang et al., 2005).

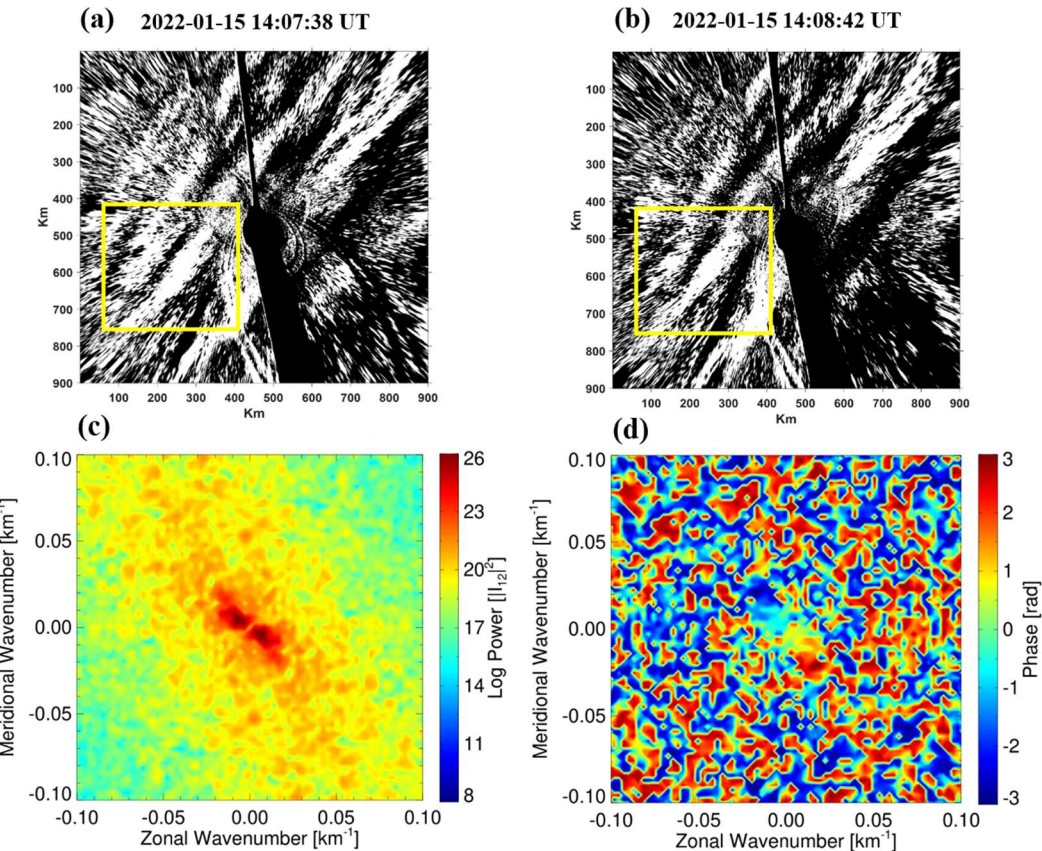

**Figure 2** The time difference images (a-b) obtained from the Xinglong OH airglow imager on the night
of 15 February 2022. Each image is projected on an area of 900 km × 900 km. The (c) cross spectrum
and (d) phase obtained from the yellow box area in the (a) and (b) using 2-D fast Fourier transform.
**2.3 Tsunami simulation model**
Tonga submarine volcano erupted on 15 January 2022, and generated tsunamis that
were detected around the globe, affected particularly the Pacific region. In this study, two
types of tsunamis were simulated, conventional tsunami simulations and atmospheric
pressure wave-induced tsunami simulations. The linear-shallow water equations in the
spherical coordinate system are used to simulate the tsunamis from the localized source and
atmospheric pressure wave. The continuity equation of a linear shallow water wave model
in spherical coordinates is:
$$\frac{\partial \eta}{\partial t} + \frac{1}{R \sin \theta} \left[ \frac{\partial (ud)}{\partial \varphi} + \sin \theta \frac{\partial (vd)}{\partial \theta} \right] = 0 \tag{1}$$

where $\eta$ is free surface elevation (m), $d$ is the water depth (m), $R$ is the Earth's
radius (6371,000 m), $\varphi$ is longitude, $\theta$ is colatitude.
While the momentum equations of the linear shallow water wave model are:
$$\frac{\partial u}{\partial t} + \frac{1}{R \sin \theta} \left[ g \frac{\partial \eta}{\partial \varphi} + \frac{1}{\rho} \frac{\partial p}{\partial \varphi} \right] + fv = 0 \tag{2}$$

$$\frac{\partial v}{\partial t} + \frac{1}{R} \left[ g \frac{\partial \eta}{\partial \theta} + \frac{1}{\rho} \frac{\partial p}{\partial \theta} \right] - fu = 0 \tag{3}$$

where, $u$ is the velocity along the lines of longitude (m/s), $v$ is the velocity along
the lines of latitude, $g$ is the gravitational acceleration (9.81 m/s$^2$ ), $p$ is the
atmospheric pressure (Pa), $\rho$ is the sea water density (1026 kg/m$^3$ ), $f$ is the Coriolis
coefficient. For the atmospheric pressure wave-induced tsunami simulation, the moving
change pressure terms as an input to tsunami simulation momentum equation. The
atmospheric pressure wave model is based on the Equation (1) in Gusman et al. (2022).
For the tsunami simulations from a localized source, a B-spline function (Koketsu and
Higashi, 1992) below is used to represent the circular water uplift source at the volcano:
$$f(x, y) = \sum_{i=0}^{3} \sum_{j=0}^{3} c_{k+i,l+j} B_{4-i}(\frac{x - x_k}{h}) B_{4-j}(\frac{y - y_l}{h}) \tag{4}$$

where $B_i(r) = \begin{cases} r^3/6, & i=1 \\ (-3r^3+3r^2+3r+1)/6, & i=2 \\ (3r^3-6r^2+4)/6, & i=3 \\ (-r^3+3r^2-3r+1)/6, & i=4 \end{cases}$ (5)
$x_k$ and $x_l$ stand for the coordinates of the knots along the x and y axes, h is the
characteristic diameter of water uplift, $r$ is the great-circle distance from the volcano
eruption center, $c_{1,1} = 1$ and the other $c_{k+i,l+j} = 0$. In this study, the modelling domain covers
the Pacific Ocean and some parts of Indian Ocean and the Caribbean with a grid size of 5
arc-min. For detailed tsunami simulation algorithms, please refer to Gusman et al. (2022).
The models for the 2022 HTHH volcanic eruption used in this study was estimated and
validated with observations at offshore DART stations around the Pacific Ocean in a
previous study (Fig. 3 and Fig. 7 of Gusman et al., 2022).
**2.4 Ray tracing method**
The following ray tracing equations (Lighthill, 1978) describes the propagation path of
AGWs.
$$\frac{dx_i}{dt} = \frac{\partial \omega}{\partial k_i} = c_{g_i}$$ (6)
$$\frac{dk_i}{dt} = -\frac{\partial \omega}{\partial x_i}$$ (7)
where $x_i$, $k_i$, $c_{g_i}$ (i=1, 2, 3), and $\omega$ are the position vector, wavenumber vector,
group speed, and intrinsic frequency, respectively.
Using the dispersion relation of acoustic gravity wave (Yeh and Liu, 1974), we can
assess the vertical propagation state of AGWs. The dispersion relation is as follows
$$m^2 = \frac{\omega^2}{c_s^2}(1-\frac{\omega_a^2}{\omega^2}) - k^2(1-\frac{\omega_b^2}{\omega^2})$$ (8)
where $m$ is the vertical wave number, $k$ is the horizontal wave number, $c_s$ the local speed of
sound, $\omega = k(c - u)$ is intrinsic frequency, $u$ is the background wind speed in the direction of
wave propagation from meteor radar observations and ERA-5 (Hersbach et al., 2020).
$\omega_a^2 = \dfrac{g}{T}\dfrac{dT}{dz} + \dfrac{\gamma g}{4H}$ is acoustic cutoff frequency, $\omega_b^2 = \dfrac{g}{T}\dfrac{dT}{dz} + \dfrac{(\gamma - 1)g}{\gamma H}$ is buoyancy frequency,
$g$ is the gravitational acceleration, and $T$ is temperature from the Sounding of the
Atmosphere using Broad band Emission Radiometry (SABER) instrument on the
Thermosphere Ionosphere Mesosphere Energetics and Dynamics (TIMED) satellite. When
$\omega > \omega_a$ or $\omega < \omega_b$, $m^2 > 0$, AGW can propagate freely, while when $\omega_b < \omega < \omega_a$, $m^2 < 0$, the wave is
evanescent.
## 3. Results and Discussion
### 3.1 Upper Atmospheric Airglow Responses to HTHH Volcanic Eruption via Lamb
Waves
Five groups of atmospheric waves (wave packets #1-5) were observed in the
mesopause region by the ground-based airglow network. Refer to this Supplement
(https://doi.org/10.5446/66190) for detailed wave propagation status. To eliminate random
disturbances, we also made videos of two days before and after the volcanic eruption
(https://av.tib.eu/series/1689). From the videos, it can be seen that the OH airglow layer
was very calm during this period. Figure 3 shows the wave packet #1 observed by the
airglow imager network (top panels). Wave packet #1 entered the view of the airglow
network approximately 8 hr after the HTHH volcanic eruption (Left image of top panels).
Three hours after wave packet #1 entered the field of view, wave packet #2 was observed
by the airglow network. The leading front of wave packet #2 has an uninterrupted

continuous front, which almost covers the whole Chinese Mainland (middle panels).

Interestingly, we observed AGWs accompanying wave packet #2 (hereafter wave packet #3)

over the northwest region of the Yellow Sea (Left image of middle panels). Wave packet #2

always keeps a stable state in the process of propagation, and maintains a regular front

when propagating over Lhasa Station (29.7ºN, 91.0ºE). Wave packet #4 exhibits strong

instability characteristics during propagation. Compared to the continuous leading front of

wave packet #2, the fronts of wave packets #4 and #5 are separated (bottom panels). We

also found that wave packet #5 propagate more than 3000 km inland (propagating to the

area west of longitude 90°E).

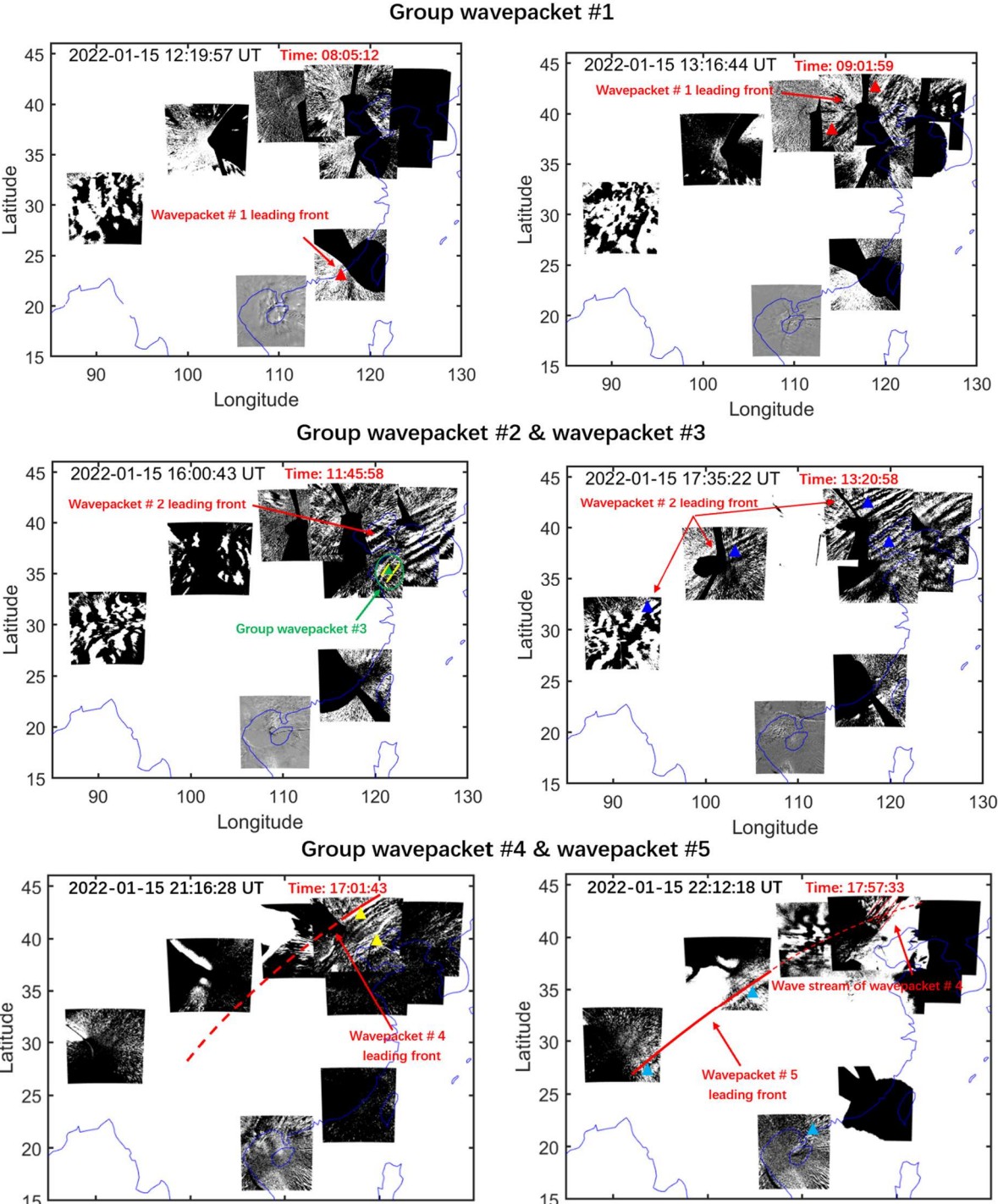

**Figure 3** Five strong group atmospheric waves associated with the Tonga volcano eruptions were observed in the mesopause region by the ground-based airglow network. Different colored triangles correspond to each wave event sampling point, while red, blue, green, yellow, and cyan correspond to wavepackets #1, #2, #3, #4, and #5, respectively. The red time markers in this figure and the following figure represent the lapse time since the volcano eruption.

Figure 4 shows the distribution of wave parameters for multi-group of atmospheric

waves (wave packets #1-#5) from cross spectral analysis. The phase speed of wave packet
#1 leading front is approximately 309 m/s. Wave packet #2 displays a slightly slower phase
speed, with average phase speed of 236 m/s. The horizontal phase speeds of group wave
packets # 3-5 are mainly distributed in the range of 200 m/s to 215 m/s, which is smaller
than that of wave packets # 1-2. The horizontal wavelengths of these five group wave
packets are mainly distributed in 80 km-105 km, while the observation periods are
relatively small and mainly concentrated in 5.7 min-7.2 min. For amplitude, the average
amplitude of the lamb wave L1 mode (5.4%) is higher than that of the lamb wave L0 mode
(3.2%). Wavepackets # 3, # 4, and # 5 have relatively small amplitudes, mainly distributed
between 0.85% and 1.25%.

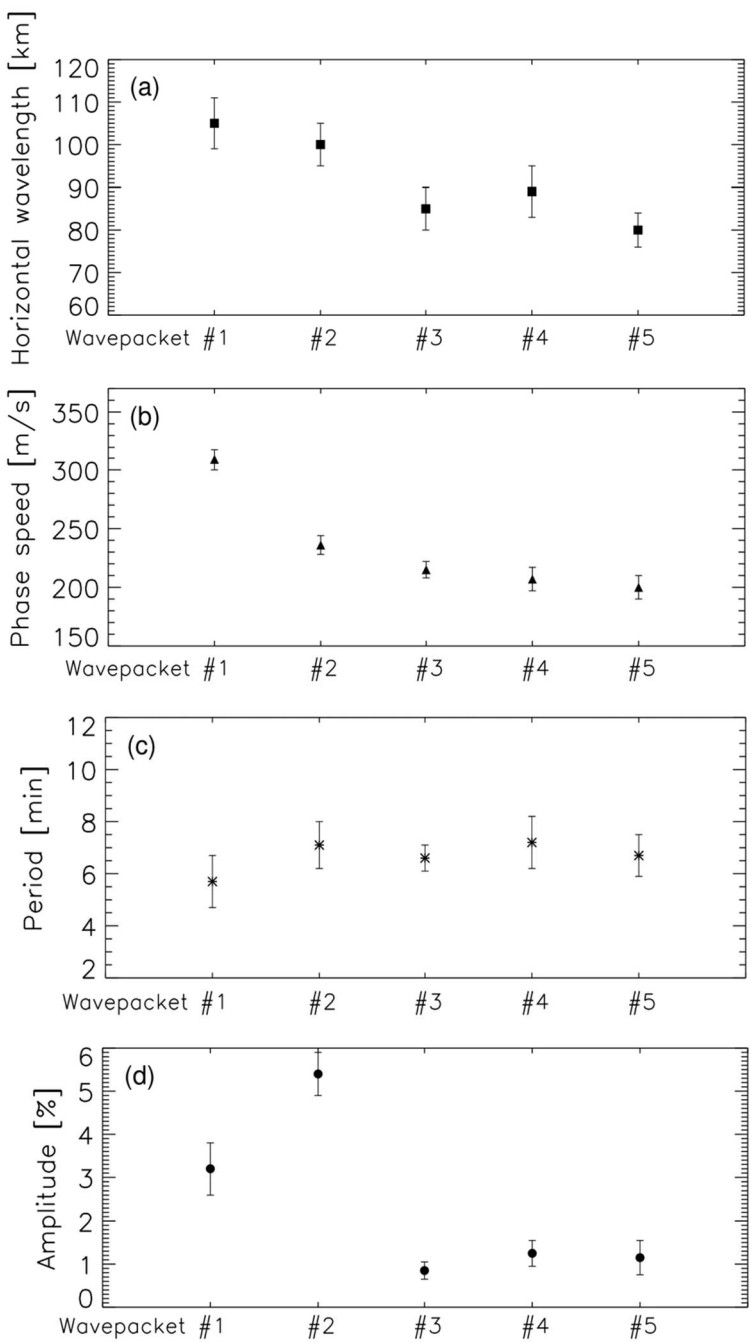


**Figure 4** Distribution of (a) horizontal wave wavelength, (b) phase speed, (c) period, and (d) amplitude parameters for multi-group of atmospheric waves (wave packets #1-#5). The calculation of wave packet parameters comes from the average value of the wave passing through the sampling points in Fig 3.

The HTHH volcano eruption produced Lamb waves that propagate around the globe, (Wright et al., 2022) causing sudden changes in surface pressure (Omira et al, 2022; Takahashi et al., 2023). Figure 5f shows the surface air pressure data of Xinglong station (40.4ºN, 117.6 ºE). At 13:15 UT on January 15, 2022, the air pressure dropped sharply from

920 Pa to 917.7 Pa, indicating that Lamb wave arrived at the surface of Xinglong station at
13:15 UT. A small disturbance of air pressure occurs at 16:33 UT. Figures 5e and 5d present
Himawari-8 6.2 μm brightness temperature at 13:10:00 UT (Otsuka, 2022). It can be seen
that the leading front of Lamb wave L0 mode happens to pass through the zenith direction
of Xinglong station. The time when wave packet #1 (Fig. 5b) and wave packet #2 (Fig. 5c)
reach the zenith direction of Xinglong Station from OH airglow observation is 13:13:34 UT
and 16:32:16 UT, which matches the time for surface pressure disturbances quite well. The
phase speed of the wave packet #1 leading front (~309 m/s) is very close to the speed of
surface Lamb wave (L0 mode). From the Fig 5, it can be seen that the phase of the lamb
wave L0 mode is almost vertical from the ground to the stratosphere and then to the
mesosphere. The wave packet # 2 with a slower phase speed (~236 m/s) is consistent with
the Lamb wave L1 mode in theoretical predictions (Francis, 1973) and simulations from
WACCM-X model (Liu et al., 2023). However, at almost the same time, the wave front
observed in the thermosphere (Video Supplement, https://doi.org/10.5446/66280) with a
slightly faster phase speed of 342 m/s is nearly 550 km a head of the wave front in the
mesopause region in the horizontal propagation direction and ahead of time approximately
30 min (Fig. 5a). This is in good agreement with theoretical and modeling results (Fig. 4 of
Lindzen and Blake, 1972; Fig. 2 of Liu et al. 2023), which show that the wave fronts of
Lamb wave below the lower thermosphere are vertical and tilt forward above. As for Lamb
wave L1 mode, the ground and mesopause region provide waveguide surfaces, resulting in
maximum wave energy between the two layer, while the phase does not change with height
(Francis, 1973).
As for why the observed Lamb wave L0 shape in the OH airglow layer is not a strong
leading wave with much weaker trailing waves, it may be caused by the following factors.
It is seen from model simulations that the wave amplitudes of L0 and L1 modes are not
uniform at the wave front. This non-uniformity becomes more pronounced in the upper
atmosphere (e.g. Fig 2 of Liu et al., 2023), probably as a result of the large variation of the
background atmosphere propagation conditions. It is thus possible that over certain regions
the trailing waves become comparable with the leading wave. It is also possible for the
leading wave to gradually dissipate energy and become invisible during propagation by
generating trailing waves. In addition, due to the smaller field of view of the airglow
imager compared to satellite observations, some structures may be related to local fine
structures, especially in the middle and upper layers where many internal waves have
significant amplitudes, which may be relatively more significant than Lamb waves.
As mentioned above, the amplitude of Lamb wave L1 mode in the mesopause region
is greater than that of L0 mode, which may be due to the fact that L1 mode is an internal
wave below the mesopause (Liu et al. 2023). For an isothermal atmosphere, the Lamb wave
L0 mode amplitude grows with altitude $z$ as $e^{\kappa z/H}$, where $H$ is the scale height, $\kappa = (\gamma - 1)/\gamma$,
and $\gamma$ is the ratio of specific heats ($\sim 1.4$). However, the amplitude of internal GWs varies as
$e^{z/2H}$. The amplitude of internal waves increases with height at a rate greater than that of
surface modes.
Poblet et al. (2023) reported observation of Lamb wave L1 mode in the horizontal
wind field of meteor radar, but they do not see Lamb waveL0 mode and argue that L0 mode
is likely a higher-frequency wave and got averaged out. Stober et al. (2018, 2024) found
that the anomalous peak signal in the meteor radar wind field cannot be completely
determined to be caused by the Lamb wave generated by the Tonga volcanic eruption. On
the one hand, meteor radar observations may have filtered out high-frequency Lamb waves.
On the other hand, even if Lamb waves are observed in the upper atmosphere, there is still
debate over whether they propagate directly to the upper atmosphere or through multi-step
vertical coupling process described by Becker and Vadas (2018), Vadas and Becker (2018),
and Vadas et al. (2018, 2023).

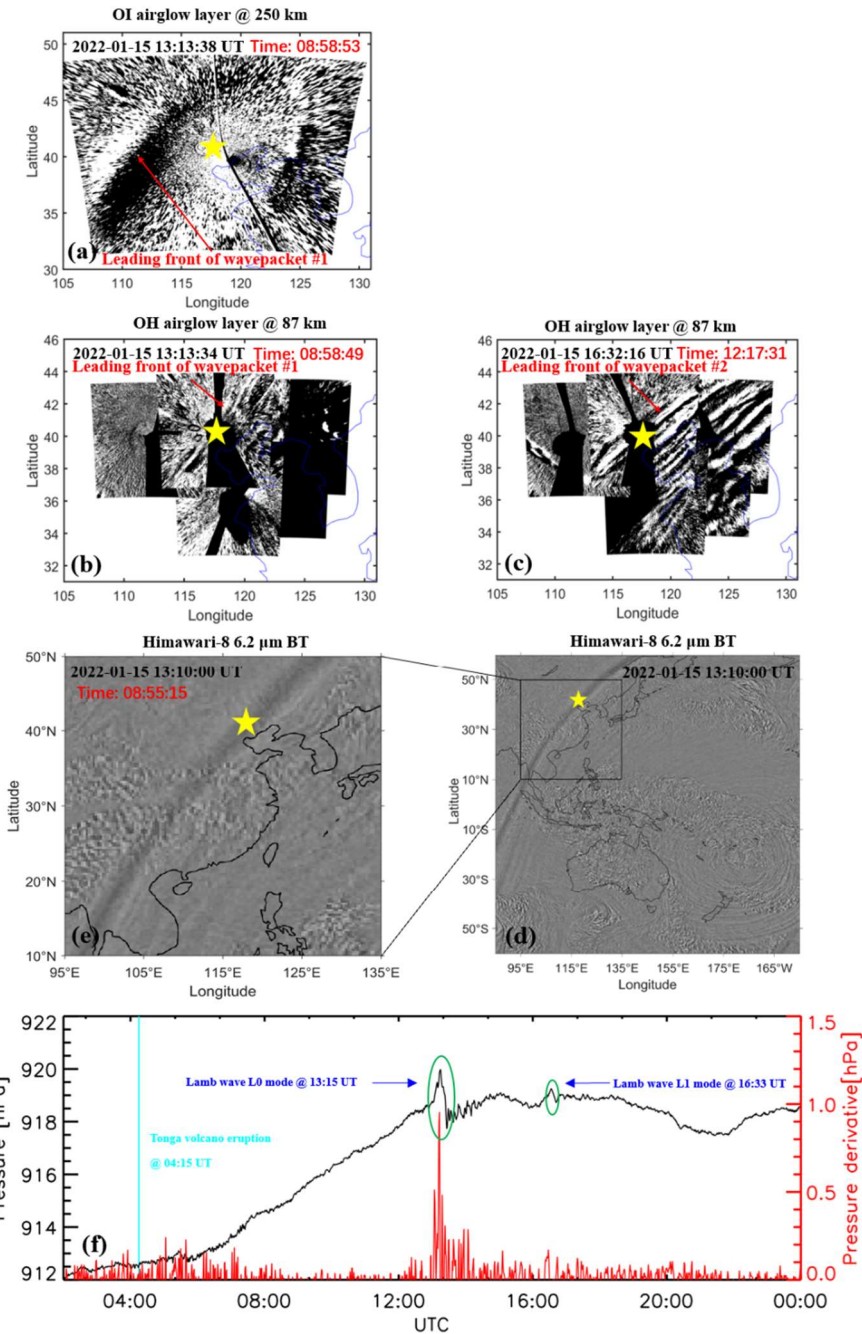

**Figure 5** (a) OI 630 nm airglow observation at 13:13:18 UT. OH airglow network observations when (b) wave packet #1 and (c) wave packet #2 pass through the zenith direction of Xinglong Station at 13:13:34 UT and at 16:32:16 UT, respectively. (d)-(e) Himawari-8 6.2 μm brightness temperature at 13:10:00 UT. (f) The surface time series of surface pressure obtained from Xinglong observation station. The red line represents the time derivative of the pressure. The sudden change of air pressure at 13:15 UT indicates the arrival time of Lamb wave L0. A small disturbance of air pressure occurs at 16:33 UT indicates the arrival time of Lamb wave L1. The yellow stars represent the location of the Xinglong station.

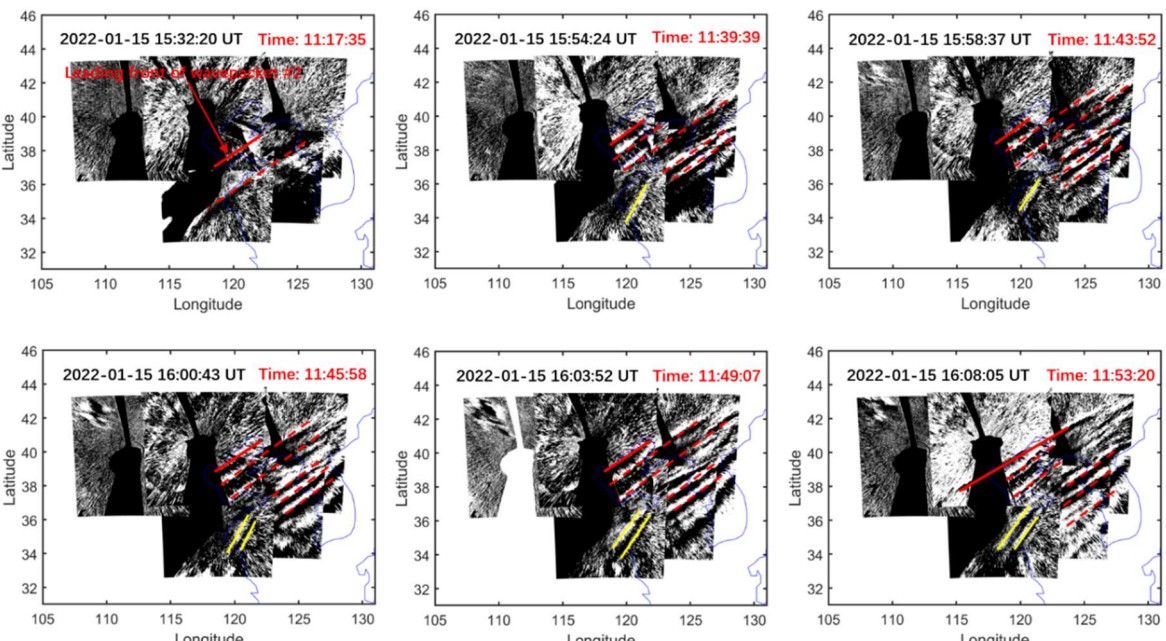

**Figure 6** The red solid lines indicate leading wave front of the wave packet #2. The yellow solid lines mark wave packet #3, which are clearly not parallel to the wave fronts of wave packet #2.

Figure 6 shows the time sequence of propagation image of wave packet #3. We found that with the propagation of wave packet #2, there is an AGW (wave packet #3) with a certain angle between its phase plane (yellow solid line) and the phase plane of wave packet #2. This implies that the source of the wave packet #3 is different from that of wave packet #2. The horizontal wavelength of the wave packet #3 near the coast is 84 km ± 5 km.

## 3.2 Simulation of Tsunami induced by HTHH Volcano Eruption

The 2022 HTHH volcano eruption triggered global atmospheric pressure waves. The simulated atmospheric pressure waves propagateat an approximate constant speed of 317 m/s, and the amplitude decreases with the distance from the volcano (Gusman et al., 2022). Figure 7 shows snapshots of the TIAPW and TITVE simulation results. The leading TIAPW excited by the pressure disturbances travels at the same speed as the atmospheric

pressure wave and is followed by subsequent sea waves generated earlier in the
atmospheric pressure wave propagation which thereafter travel at the conventional tsunami
propagation speed. Under a given pressure gradient, the discharge flux in deep sea is much
greater than that in shallow water. A deep bathymetric feature such as the Kermadec Tonga
Trench can more effectively generate tsunami waves. The wave train following the leading
wave travelling over the trench appear to be larger than those travelling in other directions.
The propagation speed of TITVE from the shallow water (long) wave approximation is
$v = \sqrt{gH_0}$ (Salmon, 2014), where g is the gravitational acceleration and $H_0$ is the ocean
depth. For sea water with a general depth of 4 km, the speed of shallow water wave is about
200 m/s. Therefore, the TIAPW is significantly faster than the TITVE. The amplitude of
TITVE is greater than that of tsunamis generated by atmospheric pressure waves. The wave
train following the leading wave of TITVE exhibit finer structures with scales smaller than
that of TIAPW. We found that the TIAPW arrived along the coast of Chinese Mainland
about 4-5 hours earlier than the TITVE.

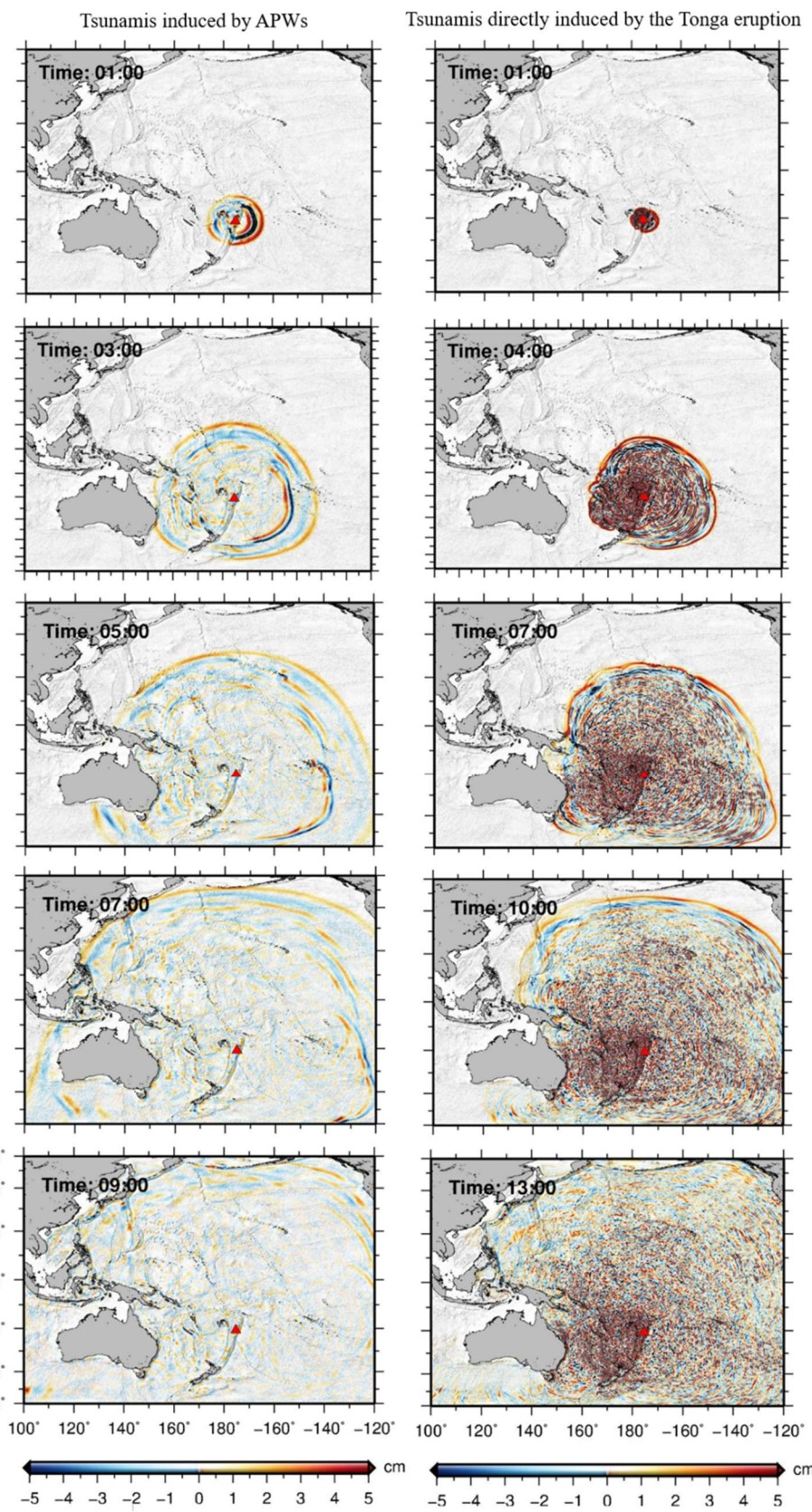


**Figure 7** Snapshots of simulated tsunamis induced by the atmospheric pressure wave (left panels) and

tsunamis directly induced by the Tonga volcano eruption (right panels).

## 3.3 Upper atmosphere responses to HTHH volcanic eruption via Air-Sea Interaction

Figure 8 shows the simulation results of TIAPW and TITVE near the coast of Chinese Mainland 11 hr (15:15 UT) and 15 hr (19:15 UT) after the volcanic eruption, respectively. Air pressure waves are not very efficient at directly exciting tsunamis in shallow water due to the weaker air-sea coupling (Gusman et al., 2022; Yamada et al., 2022). The Yellow sea is quite shallow, so the amplitude of the leading of TIAPW is very small there. The leading wave is followed by subsequent waves with larger amplitudes, which propagate in the same direction as the leading wave but at the conventional tsunami speed (Gusman et al., 2022). We found that the TIAPW and TITVE on the continental shelf have shorter wavelengths compared with those in the deep ocean. When the tsunamis approached the coast of China, three groups of AGWs (wave packet #3 and wave packets #4-5) were observed by the airglow network. The time when the AGW entered the view of the airglow network was very close to the time when the Tonga tsunamis reached the coast of Chinese Mainland. The wave packet #3 entered the airglow network at 15:30 UT and the wave packets #4-5 entered the airglow network at 19:40 UT. This strongly suggests that the wave packets detected by the airglow network are correlated to the tsunamis near the coast. We found that as the tsunamis approached the coast of China, they diffracted between Taiwan and Philippines and became discontinuous. And the wave packets #4 and #5 we observed was also discontinuous, which further confirms the correlation between wave packets # 4-5 and discontinuous tsunamis. We estimate that the average wavelength of TIAPW near the coast of the Yellow Sea is approximately 82 km ± 4 km, which is very consistent with the horizontal wavelengths of the atmospheric AGW observed by airglow network as mention

above (84 km ± 5 km), while the average wavelengths of TITVE near the coast of the
Yellow Sea and South Sea are 95 ± 5 km and 86 ± 5 km, respectively.

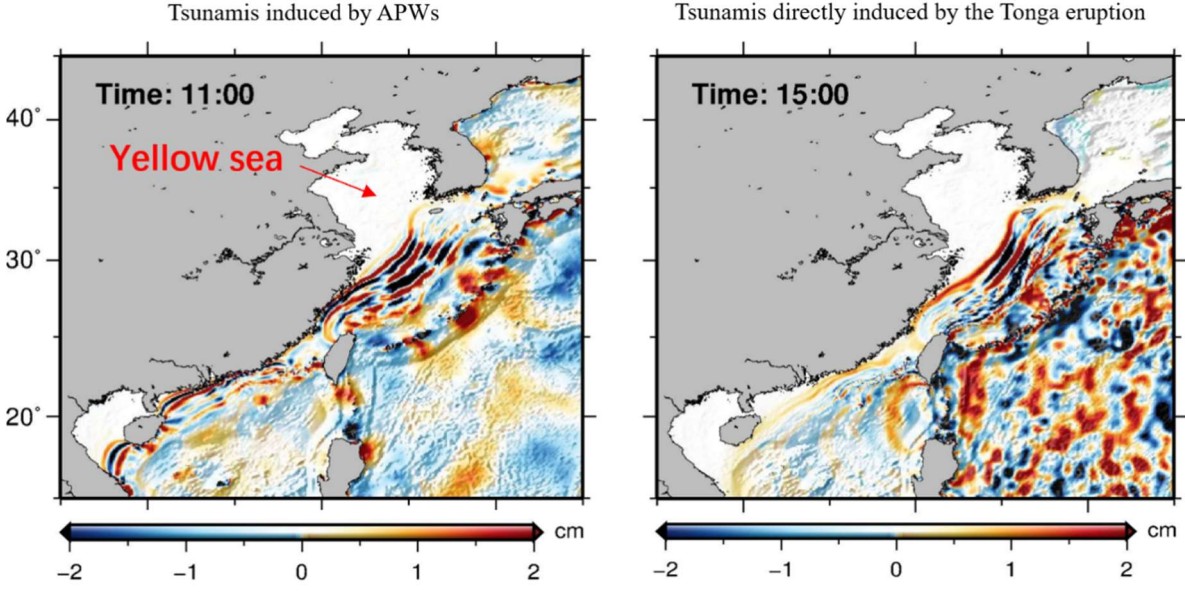


**Figure 8** Simulated tsunamis induced by the atmospheric pressure wave (left panels) and tsunamis
directly induced by the Tonga volcano eruption (right panels) near the coast of Chinese Mainland. The
marked time represents the time after the volcanic eruption.
Figure 9a shows three TIMED satellite tracks with descending track #1 along the coast
of China, ascending track #1 located east of the Korean Peninsula, and ascending track #2
inland China. Figure 9b shows the square of vertical wave number $m^2$ profile (black)
derived from the average temperature from the limb viewing of the Sounding of the
Atmosphere using SABER/ TIMED measurement locations marked by the red circles and
triangles in Fig. 9a. We take the average temperature of ascending track #1 and descending
track #1 serves as the background temperature for the wave packet #3 and ascending track
#1 as the background temperature of the wave packets #4-5 when they propagate in the
coastal vicinity. We take ascending track #2 as the background temperature of wave packets
#4-5 when they propagate inland China. The peak height of OH airglow layer is 87 km. We
found that the propagation of wave packet #3 (dash-dotted line) is in a state of free
propagation in the coastal vicinity.

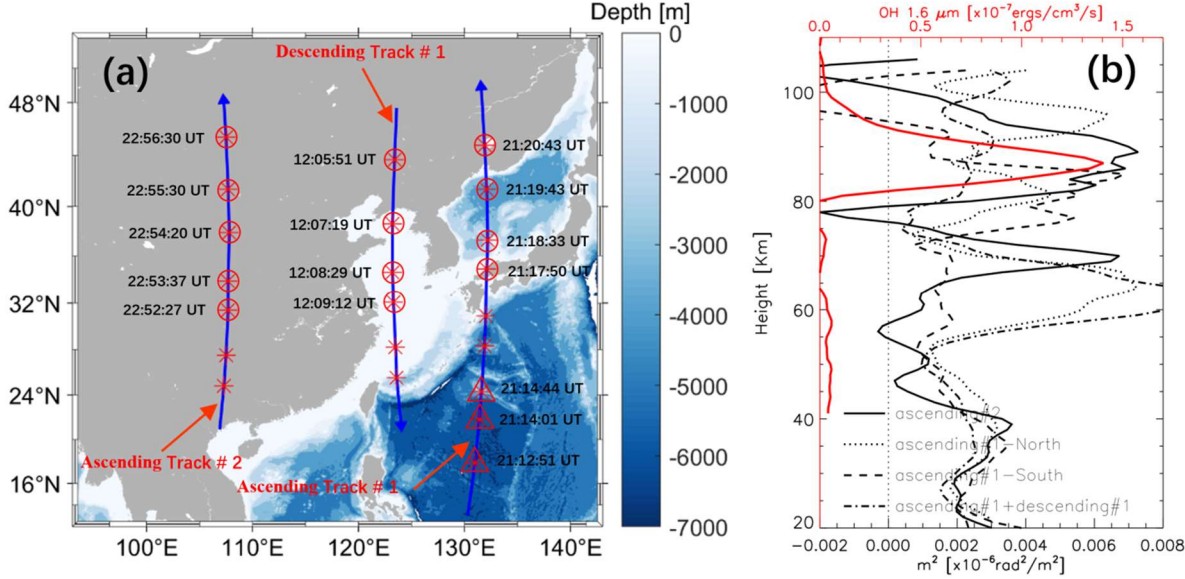

**Figure 9** (a) Ascending and descending SABER/TIMED satellite tracks over Chinese Mainland.
Background representative ocean depth map. (b) Square of vertical wave number $m^2$ profiles: black solid
line profile derived from the ascending track #2 (marked by the red circle), dotted line profile derived
from the ascending track #1-North (marked by the red circle), dashed line profile derived from the
ascending track #1-South (marked by the red triangle), and dash-dotted line profile derived from the
average the ascending track #1 and descending track #1 (marked by the red circle) from the
SABER/TIMED measurement locations in (a). The red line represents the OH 1.6 μm emission intensity
obtained by the SABER/TIMED.
Figure 10 show the background field used for ray tracing analysis for the TIAPW
event. The temperature comes from TIMED/SABER and ERA-5 and wind data from
meteor radar and ERA-5. Meteor radar wind field is from Beijing station (40.3°N, 116.2°E).
Figure 11 shows the results of ray tracing for the wave packet #3. We find that the source
location of AGWs over the coast of Chinese Mainland falls in the near coast where the
tsunami occurred.
Tsunami simulation shows that the surface wave height along the coast of Chinese
Mainland is in the order of 2 cm. There have been theoretical (Peltier and Hines, 1976) and
observational (Grave and Makela, 2015, 2017) studies on the relationship between the
amplitude of tsunamis and GWs. Peltier and Hines (1976) found that a tsunami amplitude
of ± 1 cm at sea level can cause vertical motion of ionospheric E layer and F layer ± 100 m.
A more direct observational evidence is that Grawe and Makela (2017) provided airglow
observation of tsunami-generated ionospheric signatures over Hawaii caused by the 16
September 2015 Illapel earthquake. They found that vertical disturbances on the sea surface
not exceeding 2 cm (Fig. 3b of Grave and Makela, 2017) can create detectable signatures in
the ionosphere (Fig. 1 of Grave and Makela, 2017). Therefore, we suggest that the waves
with larger amplitudes following the leading of TIAPW interact with the atmosphere after
arriving at the coast of Chinese Mainland to generate the upward propagating AGW packet.

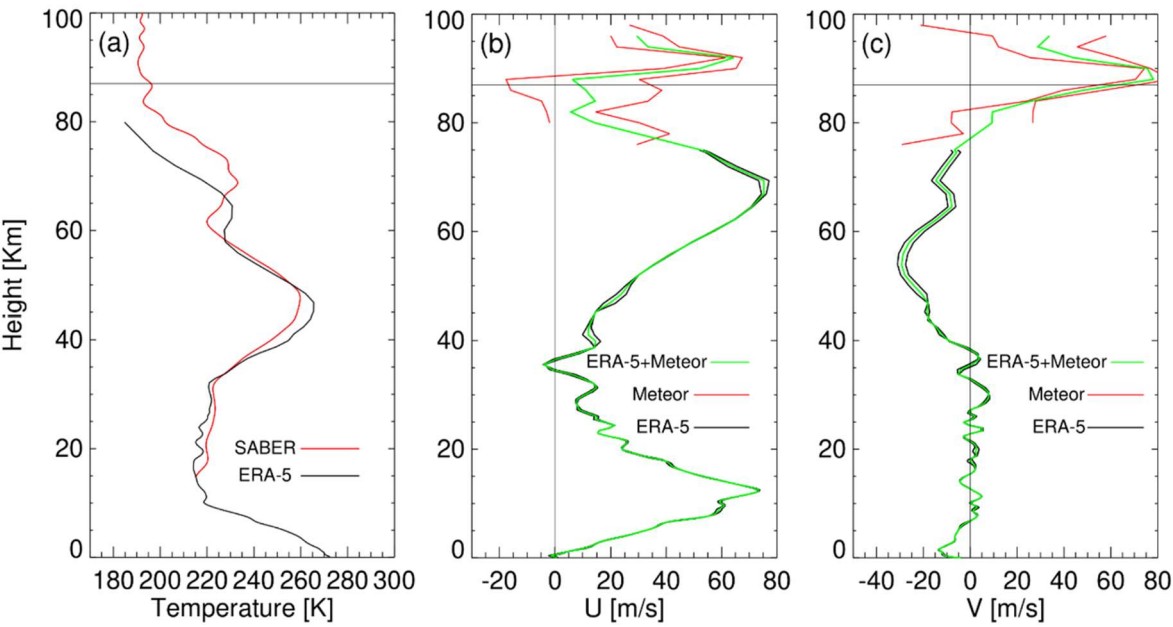

**Figure 10** The background field used for ray tracing analysis for the TIAPW event (a) Saber temperature
(red) comes from the average temperature of ascending track #1 and descending track #1 in Fig. 9, and
ERA-5 temperature (black) comes from the average of 15:00 UT and 16:00 UT. (b) Meteor zonal wind
field (red) and ERA-5 zonal wind field (black). (c) Meteor meridional wind field (red) and ERA-5
meridional wind field (black). The two red and black lines in (b) and (c) are respectively from 15:00 UT
and 16:00 UT. The green lines represent the average of two lines. Meteor radar wind field is from
Beijing station.

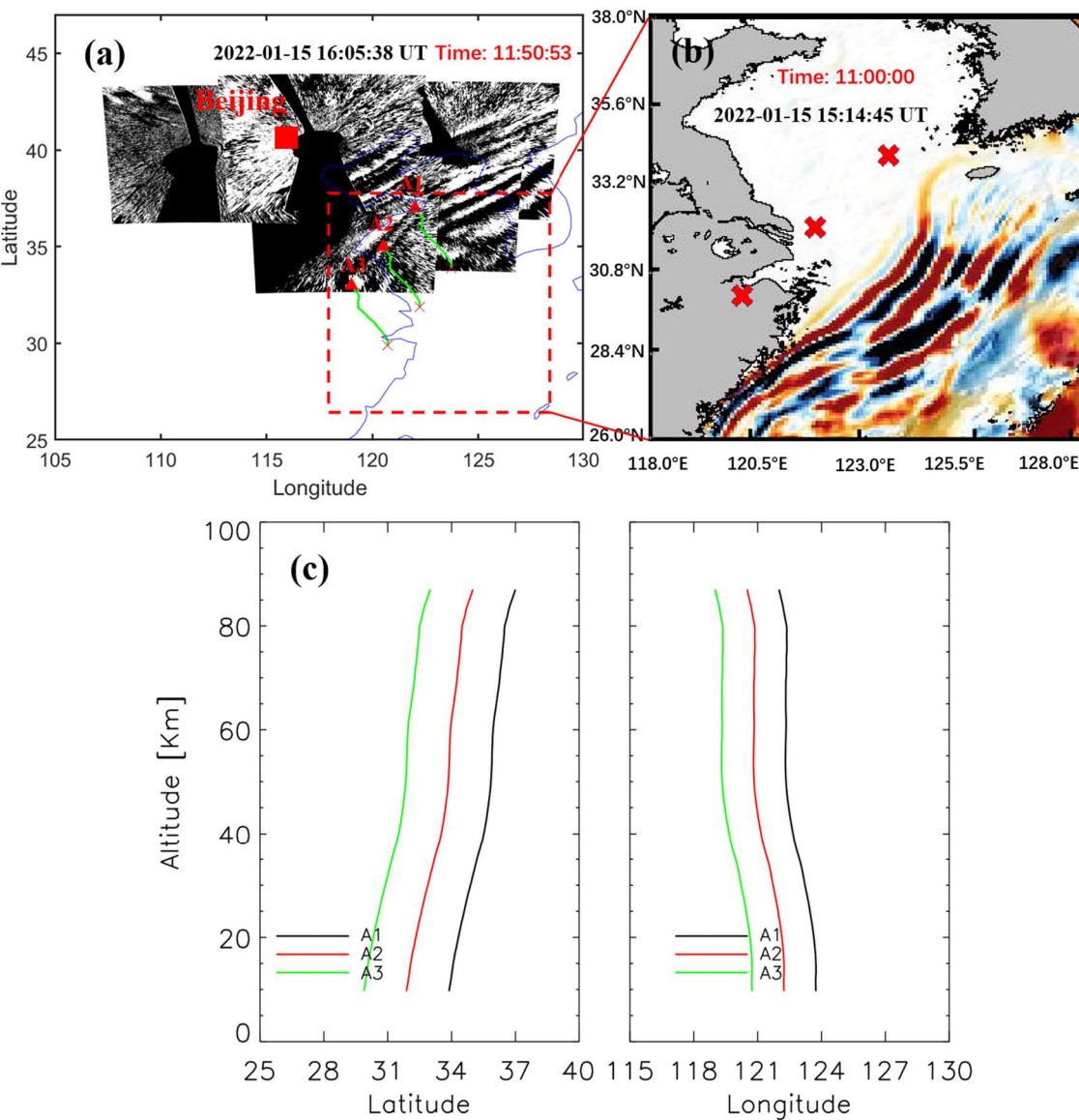

**Figure 11** (a) Backward ray tracing results of the wave packet #3 observed by the OH airglow network. The red triangles and red crosses represent the trace start and termination points, respectively. (b) Simulated tsunamis induced by the atmospheric pressure wave (TIAPW) corresponding to the dotted rectangular area in (a). (c) Ray paths of the wave starting from the seven sampling points in (a).

According to the theory of AGW dispersion, the AGW propagating obliquely has the following approximate relationship: $\sin(\varphi) \sim T_B/T$, $\varphi$ is the oblique propagation angle, $T_B$ is the buoyancy period, $T$ is the intrinsic period. Azeem et al. (2007) found that the disturbances in the ionosphere excited by the 2011 Tohoku tsunamis when they reached the west coast of the United States. They concluded that the fluctuations observed in TEC satisfy AGW dispersion relation, and the period and horizontal wavelength of the TEC

disturbances increased with distance from the West Coast of the U.S.
From the airglow network observations, we found that the wave packets #4-5 excited
by the tsunamis, continues to propagate over the main land more than 3000 km from the
coast. If the AGWs observed by the airglow network propagate freely rather than being
constrained by duct, we will obtain the propagation characteristics similar to that observed
by Azeem et al. (2007) in the ionosphere from TEC observations. $T_B$ is about 5min from
the SABER/TIMED observation. The period of wave packet #3 is between 5.5 min and 8.5
min. The minimum propagation angle $\varphi$ equals $35°$, and the corresponding maximum
propagation distance $L$ is 125 km from $L \sim H_{oh}/\tan(\varphi)$ estimation, where $H_{oh}$=87 km is the
height of OH airglow layer. However, our observation does not satisfy the free oblique
propagation dispersion theory of AGWs. In addition, we did not find that the GW
horizontal wavelength increased with the distance from the shore, as predicted by the
theory of AGW oblique propagation. Therefore, the AGWs excited by the tsunami we
observed in the mesopause region may be modulated by duct.
We did find a duct structure between 80 and 93 km (black solid line in Fig. 9b), while
the wave packet #3 were in a state of free propagation when they propagate around the
coastal vicinity of Chinese Mainland (dotted line and dashed line). The duct almost includes
the whole OH airglow layer. Therefore, we believe that AGWs generated by TITVE may
enter the duct in the process of propagation over Chinese Mainland. The duct structure over
Chinese Mainland can explain that the GWs generated by the tsunamis can propagate
thousands of kilometers inland.

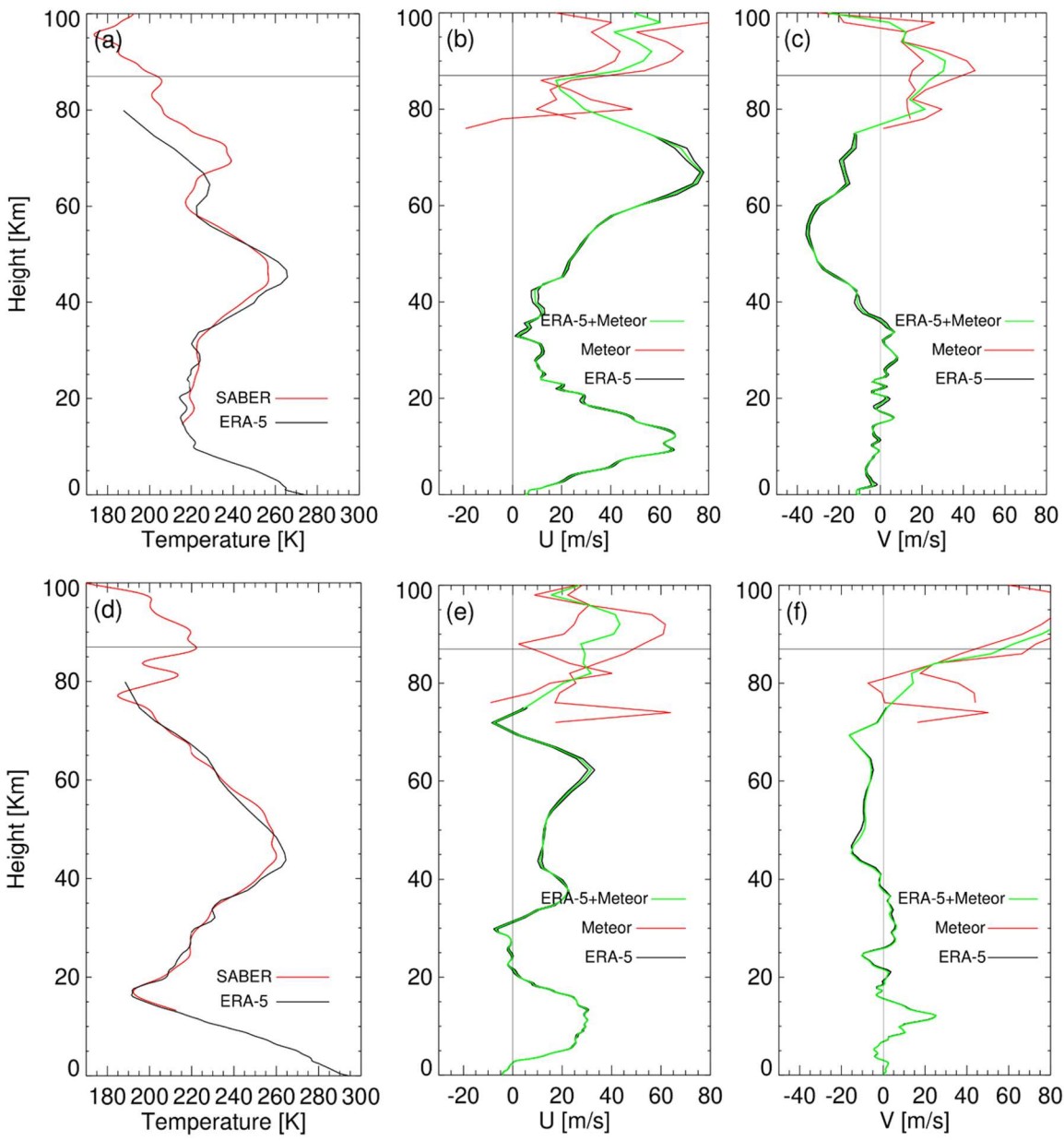

**Figure 12** Similar for Figure 10, but for ray tracing analysis for the TITVE events. The SABER temperature field in (a) comes from ascending track #1(21:17:50 UT, 21:18:33UT, 21:19:43 UT, and 21:20:43 UT) in Fig. 9, and the meteor radar wind fields in (b) and (c) come from Beijing station. The SABER temperature field in (d) is from ascending track #1 (21:12:51 UT, 21:14:01 UT, and 21:14:44 UT) in Fig. 9, and the meteor radar wind fields in (e) and (f) are from Ledong station.

Figure 13 shows the results of ray tracing for wave packets #4-5. The background field used for ray tracing analysis for the wave packets #4-5 is from Fig. 12. Meteor radar wind field is from Ledong station (18.3°N, 109.4°E). The horizontal wavelength of wave packets #4 and #5 observed near the coast by the OH airglow network approximately 89 km ± 6 km

and 80 km ± 4 km. We find that the source location of AGWs over the coast of Chinese

Mainland falls in the near tsunami area, while the location of AGW ray termination over

the inland is around 80 km (position B6 and B7 in Fig. 13d), which indicates that the wave

meets the evanescent layer (Wrasse et al., 2006). This is consistent with the duct structure

obtained through dispersion relation. Therefore, we suggest that TITVE interact with the

atmosphere after arriving at the coast of Chinese Mainland to generate the upward

propagating AGW packet. After reaching the mesopause region, this wave packet enters the

wave duct structure in the horizontal propagation process, and this wave duct supports

wave packet #5 to propagate more than 3000 km inland China.

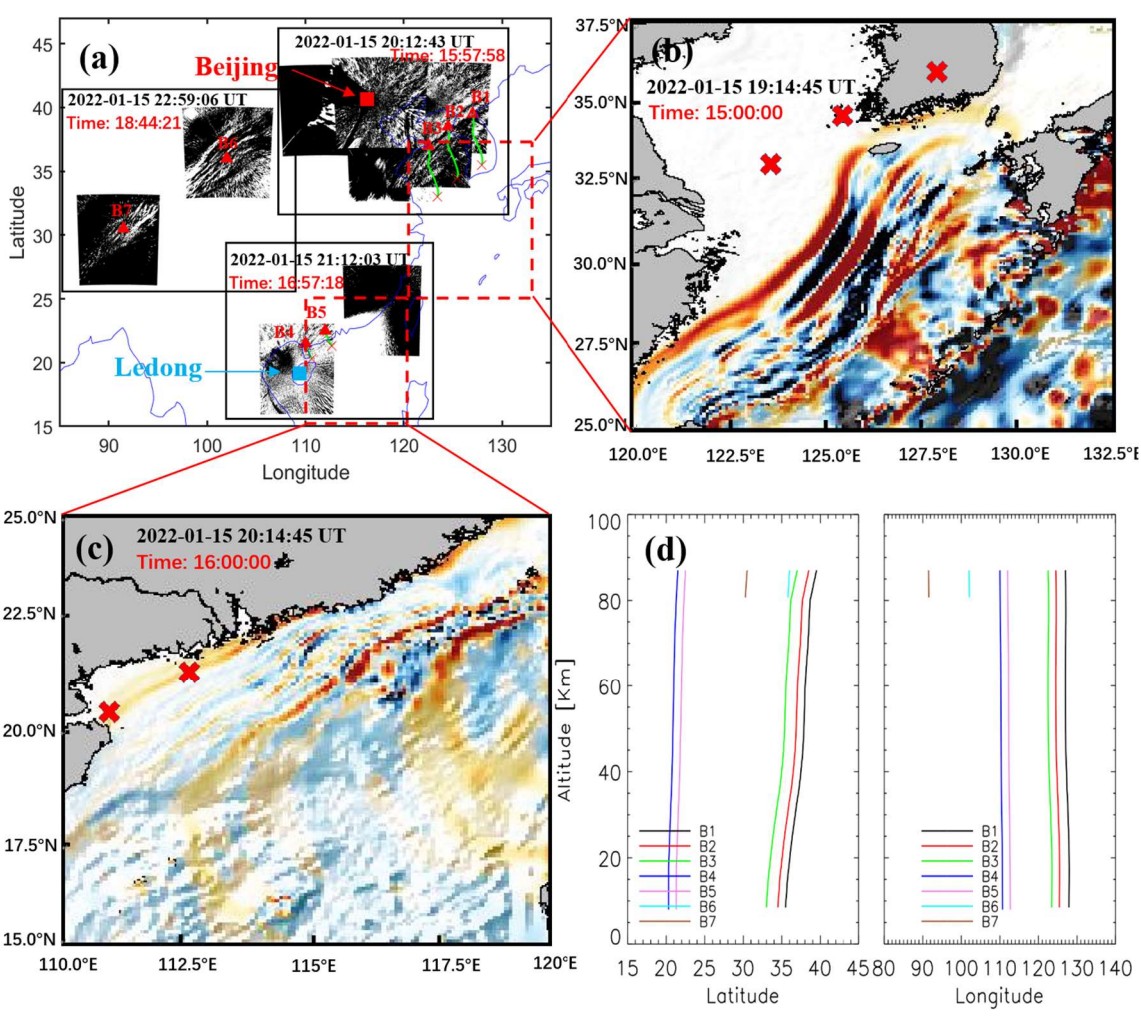

**Figure 13** (a) Backward ray tracing results of the fourth and five group GWs observed by the OH airglow

network. The red triangles and red crosses represent the trace start and termination points, respectively. (b) and (c) Simulated tsunami directly induced by the Tonga volcano eruption (TITVE) corresponding to the dotted rectangular area in (a). (c) Ray paths of the wave starting from the seven sampling points in (a).

## 4. Conclusions

Strong atmospheric disturbances, including Lamb waves, acoustic waves, and gravity waves, were triggered by the 2022 HTHH volcano eruption. The HTHH submarine volcanic eruption also triggered an unusual tsunami, which can generate atmospheric gravity waves (Fig. 14). We observed five strong group atmospheric waves associated with the HTHH volcano eruption from the ground-based airglow network observations.

The phase speed of the wave packet #1 leading front is approximately 309 m/s, which is observed almost simultaneously with the surface Lamb wave L0 mode. The high-frequency wave trains following the wave packet #1 leading front observed by the northern OH airglow imager network may also be related to the dissipation of the leading waves. Wave packet #2, with average phase speed of 236 m/s, may be considered as Lamb wave L1 mode, which exhibits internal GW behavior. Wave packet # 3 and wave packets #4-5 are generated by TIAPW and TITVE from backward ray tracing analysis. The horizontal phase speed distribution range of wave packets #3-5 is 200 m/s to 215 m/s, which is smaller than that of wave packets # 1-2.  For amplitude, the average amplitude of the lamb wave L1 mode (5.4%) is higher than that of the lamb wave L0 mode (3.2%), while wavepacket # 3, # 4, and # 5 have relatively small amplitudes, mainly distributed between 0.85% and 1.25%. The horizontal wavelengths of the atmospheric AGWs observed by the airglow network are very consistent with those of the tsunami near the coast. This is the first time that we observed the AGWs in the mesopause region triggered

by the tsunamis using optical detection equipment. It is also the first time to report
atmospheric gravity waves excited by TIAPW.

When the wave excited by TITVE propagate far away from the coast, the

characteristics of AGWs are not consistent with the dispersion of free propagation AGWs.
We find these wave packets are controlled by the duct, which can support the propagation
of these GWs for thousands of kilometers after the tsunami were stopped at the coast.
Therefore, tsunamis can have a significant impact on the upper atmosphere over inland
areas far from the ocean through AGWs.

The 2022 HTHH volcano eruption form a complex coupling relationship in the land-

ocean-atmosphere system (Fig. 14). Firstly, the heat released by the eruption has a direct
impact on the ocean, causing temperature changes in the surrounding waters. This can lead
to changes in the marine environment, affecting the behavior, distribution, and ecosystem
structure of organisms.

Meanwhile, volcanoes release gases such as carbon dioxide and sulfur dioxide.

Carbon dioxide is one of the greenhouse gases that can cause an increase in Earth's
temperature, leading to global warming. Sulfur dioxide can cause sulfuric acid mist in the
atmosphere, which affects the reflectivity and temperature of the atmosphere, and thus
affects the global climate.

Moreover, the 2022 HTHH volcano eruptions also trigger atmospheric waves and

tsunamis. The surface atmospheric pressure wave generated by the 2022 HTHH volcano
eruption can affect the upper atmosphere. The conventional tsunami triggered by the Tonga
volcano generated AGWs. The atmospheric pressure wave from the eruption generated a
fast tsunami never before observed by tsunami observation networks. When the tsunamis
reach the coast, their speeds decrease but their amplitudes increase, and the AGWs
generated by them will also affect the upper atmosphere. These AGWs play an important
coupling role between the ocean and the atmosphere by affecting the density and pressure
distribution of the atmosphere during propagation, leading to changes in the wind field and
affecting global atmospheric circulation. This study exhibits special dynamic coupling
process between air and sea via acoustic gravity waves (Fig. 14). This indirect impact on
the upper atmosphere provides a new perspective for us to study the coupling between the
ocean and the atmosphere and a key opportunity to improve the air-sea coupling model,
thereby enhancing our future ability to make tsunami warning forecasts.

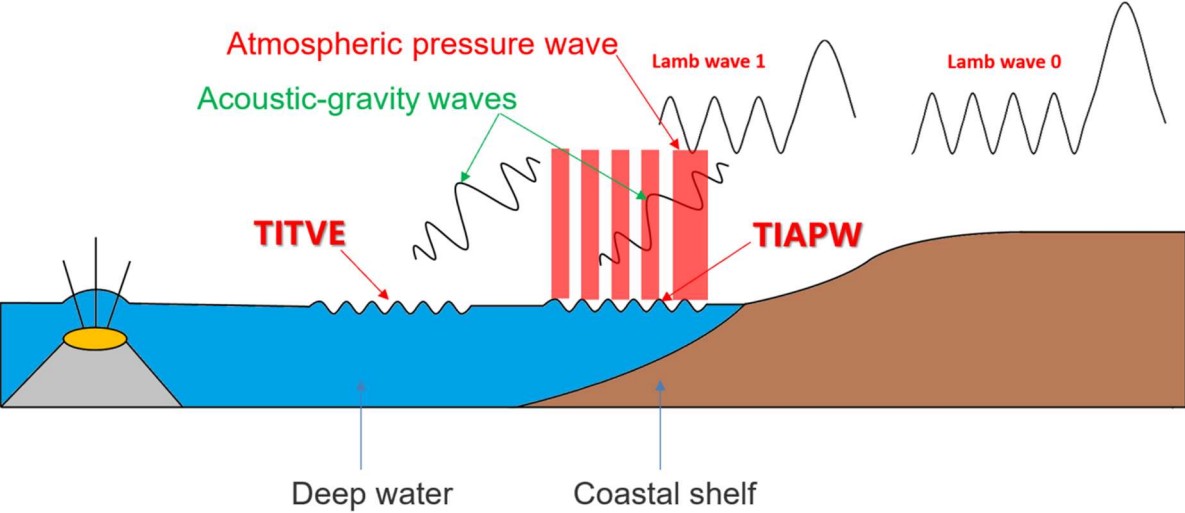


**Figure 14** The Tonga volcano eruptions triggered two types of tsunamis, one type of tsunami is induced
by the atmospheric pressure wave (TIAPW) and the other type tsunami is directly induced by the Tonga
volcano eruption (TITVE). The acoustic gravity waves (AGWs) caused by tsunamis can propagate to the
mesopause region.

**Data availability**
The Multi-Layer Airglow Network data is available
at https://data2.meridianproject.ac.cn/data (MPDC, 2024). TIMED/SABER data is accessed
from http://saber.gats-inc.com/data.php (last access: 10 January 2024). The ERA5 reanalysis
data are able to be downloaded from the Copernicus Climate Change Service Climate Data
Store   through https://www.ecmwf.int/en/forecasts/datasets/reanalysis-datasets/era5   (last
access: 12 January 2024). Himawari-8 data are distributed by the Center for Environmental
Remote Sensing (http://www.cr.chiba-u.jp/databases/GEO/H8_9/FD/index_en_V20190123.
html) (last access: 20 January 2024). Meteor data were provided by Beijing National
Observatory of Space Environment, Institute of Geology and Geophysics Chinese
Academy of Sciences through the Geophysics center, National Earth System Science Data
Center (http://wdc.geophys.ac.cn) (last access: 15 January 2024).

**Video supplements**
Multi-group of strong atmospheric waves observed over China associated with the 2022
Hunga Tonga–Hunga Ha'apai volcano eruptions (https://doi.org/10.5446/66190 Li, 2024).
Animation series of OH airglow disturbances associated with the 2022 Hunga
Tonga–Hunga Ha'apai volcano eruptions (https://doi.org/10.5446/s1689 Li, 2024). A strong
wave front observed by an OI 630 nm airglow imager over China associated with the 2022
Hunga Tonga–Hunga Ha'apai volcano eruptions (https://doi.org/10.5446/66280 Li, 2024).

**Author contributions**
J.X and Q.L. conceived the idea of the manuscript. Q.L. carried out the data analysis,
interpretation and manuscript preparation. A.R.G. developed and performed the numerical
simulations. W.L and Y.Z compiled, processed and analysed satellite data. H.L.L., X.L,

and W.Y. contributed to the data interpretation and manuscript preparation. All authors

discussed the results and commented on the manuscript.

**Competing interests**

The authors declare no competing interests.

**Acknowledgements**

This work was supported by the National Science Foundation of China (42374205 and 41974179). The project is also supported by the Specialized Research Fund for State Key Laboratories. We acknowledge the use of data from the Chinese Meridian Project.

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
