# Peer review of "Upper Atmosphere Responses to the 2022 Hunga Tonga-Hunga Ha'apai"

_EGUsphere, 2023_

## Author Comment (AC1)

**Response to Referee #1**

In the current manuscript, Li et al. have used the very high temporal and spatial resolution of the ground-based OH and OI airglow network to look at the upper atmosphere response to the HTHH (Hunga Tonga-Hunga Ha'apai) Volcanic Eruption occurred in 2022, which has been an active and important research topic after the eruption. OH airglow (? In the Figure 1) measurements have detected five group atmospheric waves in the ground-based airglow network (the first wave packet is observed after 8 hours). Then they have obtained the phase speed and amplitude over 10 hours period (?). The wave front can be also observed by OI airglow at higher altitude ~250 km. The authors also noticed that short period of sudden surface pressure changes over Xinglong station, which were caused by the pressure disturbances that spread out in the form of Lamb waves due to HTHH eruption. The authors also carried out tsunami model simulations using two sources (one is the localized source, termed TITVE, the other considers the atmospheric pressure changes, termed as TIAPW) to investigate the propagating speed to China due to different tsunami sources used in the model. Then they discussed the air-sea interactive process on the upper atmospheric responses to HTHH eruption. They also did some related important diagnosis about the vertical wavelength and ray path of the waves etc. Finally they provided a nice figure describing the dynamic coupling process between air and sea via acoustic gravity waves. Overall, the paper has a clear structure and the motivation is clear. This study is important and interesting.

However, sometimes it is unclear to me because the figures are not clear enough or some descriptions/explanations are vague (see the detailed comments below). It requires some clarification.

Thank you very much for your thoughtful and constructive comments concerning our manuscript entitled "Upper Atmosphere Responses to the 2022 Hunga Tonga-Hunga Ha'apai Volcanic Eruption via Acoustic-Gravity Waves and Air-Sea Interaction". Those comments are all valuable and very helpful for revising and improving our paper, as well as the important guiding significance to our researches. We have studied comments carefully and have made corrections which we hope meet with approval. The detailed point-by-point responses are given below.

I have two major concerns:

One is the tsunami model simulations by assuming two sources. I can not see any information about the model. Is that is a global model or regional model? What are the necessary/important input required to model tsunami? How is the model performances to represent the tsunami due to HTHH volcanic eruption?

Response:

Thank you very much for your comments. We have provided a more detailed description of the tsunami model simulations

**2.2 Tsunami simulation model**

Tonga submarine volcano erupted on 15 January 2022, and generated tsunamis that were detected around the globe, affected particularly the Pacific region. In this study, two types of tsunamis were simulated, conventional tsunami simulations and atmospheric pressure wave-induced tsunami simulations. The linear-shallow water equations in the spherical coordinate system are used to simulate the tsunamis from the localized source and atmospheric pressure wave. The continuity equation of a linear shallow water wave model in spherical coordinates is:

$$\frac{\partial \eta}{\partial t} + \frac{1}{R \sin \theta} \left[ \frac{\partial (ud)}{\partial \varphi} + \sin \theta \frac{\partial (vd)}{\partial \theta} \right] = 0 \tag{1}$$

where $\eta$ is free surface elevation (m), $d$ is the water depth (m), $R$ is the Earth's radius (6371,000 m), $\varphi$ is longitude, $\theta$ is colatitude.

While the momentum equations of the linear shallow water wave model are:

$$\frac{\partial u}{\partial t} + \frac{1}{R \sin \theta} \left[ g \frac{\partial \eta}{\partial \varphi} + \frac{1}{\rho} \frac{\partial p}{\partial \varphi} \right] + fv = 0 \tag{2}$$

$$\frac{\partial v}{\partial t} + \frac{1}{R} \left[ g \frac{\partial \eta}{\partial \theta} + \frac{1}{\rho} \frac{\partial p}{\partial \theta} \right] - fu = 0 \tag{3}$$

where, $u$ is the velocity along the lines of longitude (m/s), $v$ is the velocity along the lines of latitude, $g$ is the gravitational acceleration (9.81 m/s$^2$ ), $p$ is the atmospheric pressure (Pa), $\rho$ is the sea water density (1026 kg/m$^3$ ), $f$ is the Coriolis coefficient. For the atmospheric pressure wave-induced tsunami simulation, the moving change pressure terms as an input to tsunami simulation momentum equation. The atmospheric pressure wave model is based on the Equation (1) in Gusman et al. (2022).

For the tsunami simulations from a localized source, a B-spline function (Koketsu and Higashi, 1992) below is used to represent the circular water uplift source at the volcano:

$$f(x, y) = \sum_{i=0}^{3} \sum_{j=0}^{3} c_{k+i, l+j} B_{4-i} \left( \frac{x - x_k}{h} \right) B_{4-j} \left( \frac{y - y_l}{h} \right) \tag{4}$$

where

$$B_i(r) = \begin{cases} r^3/6, & i=1 \\ (-3r^3 + 3r^2 + 3r + 1)/6, & i=2 \\ (3r^3 - 6r^2 + 4)/6, & i=3 \\ (-r^3 + 3r^2 - 3r + 1)/6, & i=4 \end{cases} \tag{5}$$

$x_k$ and $x_l$ stand for the coordinates of the knots along the x and y axes, h is the characteristic diameter of water uplift, $r$ is the great-circle distance from the volcano eruption center, $c_{1,1} = 1$ and the other $c_{k+i,l+j} = 0$. In this study, the modelling domain covers the Pacific Ocean and some parts of Indian Ocean and the Caribbean with a grid size of 5 arc-min. For detailed tsunami simulation algorithms, please refer to Gusman et al. (2022).

**References**

Gusman, A.R., Roger, J., Noble, C. et al. The 2022 Hunga Tonga-Hunga Ha'apai Volcano Air-Wave Generated Tsunami, Pure and Applied Geophysics, 179, 3511–3525, https://doi.org/10.1007/s00024-022-03154-1, 2022.

Koketsu K. and Higashi S.: Three-dimensional topography of the sediment/basement interface in the Tokyo Metropolitan area, Central Japan, Bull. seism. Soc. Am., 82, 2328-2349, https://doi.org/10.1785/BSSA0820062328, 1992.

The other is about the inconsistent use of meteorological data product when the authors did some diagnosis (for example vertical wave numbers, lines 120-125). The authors use temperature profiles from SABER (20-100km) and the wind from both ERA5 and HWM-14. I know ERA5 is only up to 85 km and the authors would like the profile of vertical wave number up to 100 km. Which altitude range use ERA5, then HWM-14. How realistic is the HWM-14 modelled wind changes (since this model is an empirical model which only uses geomagnetic Ap index) due to the HTHH volcanic eruption? What will be the difference when you use both ERA5 wind and temperature profiles below some certain altitude compared with the current result?

Response:

Thank you for your criticism and comments. Yes, you are right. The HWM wind field is an empirical model wind field that can only reflect a long-term trend, which is not suitable for event research. Therefore, we use meteor radar wind fields to replace the HWM wind fields. The ERA-5 wind field is used in the areas below the mesopause region. Here, we use a total of 137 layers of ERA-5 wind field, with heights from the ground to 80 km. The upper boundary area may have errors due to sponge layer effects.

The height range of ERA-5 wind field used is from 0 to 75 km, and above 75 km, meteor radar wind field is used. We have reconfirmed the location of wave sources in ray tracing analysis using new wind field data.

[Figure]

**Figure 10** The background field used for ray tracing analysis for the TIAPW event (a) Saber temperature (red) comes from the average temperature of ascending track #1 and descending track #1 in Fig. 9, and ERA-5 temperature (black) comes from the average of 15:00 UT and 16:00 UT. (b) Meteor zonal wind field (red) and ERA-5 zonal wind field (black). (c) Meteor meridional wind field (red) and ERA-5 meridional wind field (black). The two red and black lines in (b) and (c) are respectively from 15:00 UT and 16:00 UT. The green lines represent the average of two lines. Meteor radar wind field is from Beijing station.

[Figure]

**Figure 11** (a) Backward ray tracing results of the wave packet #3 observed by the OH airglow network. The red triangles and red crosses represent the trace start and termination points, respectively. (b) Simulated tsunamis induced by the atmospheric pressure wave (TIAPW) corresponding to the dotted rectangular area in (a). (c) Ray paths of the wave starting from the seven sampling points in (a).

[Figure]

**Figure 12** Similar for Figure 10, but for ray tracing analysis for the TITVE events. The SABER temperature field in (a) comes from ascending track #1 (21:17:50 UT, 21:18:33UT, 21:19:43 UT, and 21:20:43 UT) in Fig. 9, and the meteor radar wind fields in (b) and (c) come from Beijing station. The SABER temperature field in (d) is from ascending track #1 (21:12:51UT, 21:14:01 UT, and 21:14:44 UT) in Fig. 9, and the meteor radar wind fields in (e) and (f) are from Ledong station.

[Figure]

**Figure 13** (a) Backward ray tracing results of the fourth and five group GWs observed by the OH airglow network. The red triangles and red crosses represent the trace start and termination points, respectively. (b) and (c) Simulated tsunami directly induced by the Tonga volcano eruption (TITVE) corresponding to the dotted rectangular area in (a). (c) Ray paths of the wave starting from the seven sampling points in (a).

Following your suggestion, taking the TIAPW event as an example, we compared the differences in ray tracing results between Saber temperature and ERA-5 temperature. Figure S1 (below) shows the deviation of ray tracing trajectories obtained from two different temperatures. From the ray tracing trajectory, there is no significant deviation between the two (with a maximum error of ± 0.32 °), and some areas almost overlap.

[Figure]

**Figure S1** Deviation of ray tracing trajectories obtained from two different temperatures

In the Introduction, the authors have introduced HTHH volcanic eruption triggered broad spectrum atmospheric disturbances including different waves (Lines 41-44), but the second paragraph mainly focuses on Lamb waves. Why are other waves ignored?

 Response:

Thank you for your suggestion.

The discussions of other waves are given below

Acoustic-gravity waves (AGWs) are mechanical waves in compressible fluids in a gravity field (Gossard and Hooke, 1975). If the frequencies are much larger than the buoyancy frequency, AGWs tend towards acoustic wave mode, and when the frequency is much smaller than the buoyancy frequency, the fluid can be considered incompressible, and the AGWs tend towards internal GWs mode. The term "acoustic-gravity waves" is usually used when restoring forces due to both gravity and compressibility are important. AGWs are known to play a significant role in the coupling between the atmosphere and the ocean (Donn and Balachandran,1981; Harkrider and Press, 1967; Press and Harkrider, 1962). Atmospheric pressure waves are mechanical waves that are related to the density of the atmosphere. Compression and expansion are the high-pressure and low-pressure regions of motion in a medium.

Specified comments:

- Line 54, it reads weird "from GNSS TEC analysis". Better to change what "analysis" method used based on "GNSS TEC". The acronym "GNSS TEC" should be described when they first appear in the text.

  Response:

  Thank you very much for your comment. We re-described this sentence as:" Li et al. (2023) identified Lamb wave L1 mode using phase-leveling amplitude technology based on global navigation satellite system (GNSS)- total electron content (TEC)." in the revised manuscript.

- Lines 58-60. Can you provide some explanation about the "atmospheric pressure wave"?

  Response:

  Atmospheric pressure waves are mechanical waves that are related to the density of the atmosphere. Compression and expansion are the high-pressure and low-pressure regions of motion in a medium.

- Line 60, AGWs is defined in the Abstract for the first time but not in the main text. Better to define it here.

  Response:

  Thank you very much for your comment.

  We define AGWs here:

  Acoustic-gravity waves (AGWs) are mechanical waves in compressible fluids in a gravity field (Gossard and Hooke, 1975).

- Line 61, please delete "the height of".

  Response:

  "the height of" is removed from the revised manuscript.

- Line 64, the authors need to make clear "that arrived before the tsunami", for example when/where Tsunami occurred, how many hours earlier etc.

  Response:

  Thank you very much for your comment. We re-described this sentence as:"Using the red line airglow imager, Makela et al. (2011) detected airglow disturbance in Hawaii that arrived 1hr earlier of the tsunami generated by the 11 March 2011 Tohoku earthquake." in the revised manuscript.

- Line 65. Again, it is very vague "sea wave and GW almost simultaneously in Chile". I guess the authors still talk about tsunami, but should make it clear if this is the same Tsunami as mentioned in Line 64.

  Response:

  Thank you very much for your comment. We re-described this sentence as:" Smith et al. (2015) observed tsunamis and GW almost simultaneously in Chile." in the revised manuscript.

- Line 66, though readers know 3D is three dimensional, better to add this after 3D.

  Response:

  "three dimensional" is added to the revised manuscript like this "three dimensional (3D)".

- Lines 69-70, I am confused with "AGWs on the mesopause airglow radiation".

  Response:

  Thank you very much for your comment. We re-described this sentence as:"Inchin et al. (2022) performed the numerical simulations of mesopause airglow radiation fluctuations induced by tsunami-generated AGWs" in the revised manuscript.

- Line 73. What is "convention tsunami"?

  Response:

  Conventional tsunamis are typically generated by localized sea surface displacements caused by sources such as earthquakes and volcanoes.

- Line 75, What is "this typical type tsunami"?

  Response:

  This typical type tsunami is convention tsunami.

- Lines 58-59, 77-78. Again, It is not clear why "tsunamis induced by the atmospheric pressure wave (TIAPW)" is more important than "tsunamis directly induced by the 2022 Tonga volcano eruption (TITVE)".

  Response:

  Thank you very much for your comment.

  We are very sorry for the confusion caused by our description. Both types of tsunamis are very important.

"Another significant mechanism that occurred was the atmospheric pressure wave that excited the tsunamis." is changed to "Another tsunami is induced by the atmospheric pressure wave (TIAPW)."

The following description is removed from the manuscript

"Neither has AGWs originate from tsunamis induced by the atmospheric pressure wave (TIAPW) been studied."

- Line 83, change "(air-water-air-coupling process)" to "through air-water-air-coupling process"?

Response:

Thank you very much for your suggestion. "(air-water-air-coupling process)" is changed to "through air-water-air-coupling process".

- Line 86, why "Double layer airglow network" since the authors mentioned "multi-layer" in Line 87?

Response:

Thank you very much for your careful comment. "double layer" is changed to "multi-layer".

- Line 93, please add the latitude/longitude information about "Xinglong Station".

Response:

The latitude/longitude information (40.4°N,117.6°E) is added after "Xinglong Station".

- Line 94-95. Are all the airglow (OH, OI, 557 nm) having the same resolution as "The temporal resolution is 1 min and the spatial 95 resolution is 1 km"?

Response:

The time resolution of OH and 557 nm airglow imager is 1 minute, while the resolution of OI airglow is 2 minutes. The spatial resolution of the airglow imager at the airglow layer is not uniform. The resolutions of OH, OI 557 nm, and OI 630 nm airglow in the zenith direction are 0.27 km, 0.29 km, and 0.77 km, respectively, while in the zenith angle of 60°, the resolutions are 1.01 km (OH), 1.11 km (OI 557 nm), and 2.65 km (OI 630 nm), respectively.

- Line 95. Can you add some reference for the "standard star map"? Or some reference how the calibration works "with the help of standard star map" here.

Response:

The following reference is added to the revised manuscript.

Garcia, F. J., Taylor, M. J., and Kelley, M. C.: Two-dimensional spectral analysis of mesospheric airglow image data, Appl. Optics,36(29), 7374–7385, 1997.

- Line 96, Again, some reference is needed for "removed by differential method".

Response:

The following reference is added to the revised manuscript.

Swenson, G. R. and Mende, S. B.: OH emission and gravity waves (including a breaking wave) in all-sky imagery from Bear Lake, UT, Geophys. Res. Lett., 21, 2239–2242, 1994.

- Sections 2.2, the model description is not clear at all! Do you use WACCM-X model which has been mentioned several times in the whole paper? This part just asks the readers to read two cited papers and has never mentioned the details what the conditions to be able to simulate the tsunami after HTHH. How readers know the model results robust and reliable? Can you explain why the second type of tsunami sources is better or more realistic than the first localised source? Readers can not judge it because there is no detailed information about the "Tsunami simulation model".

Response:

Thank you very much for your comment.

We did not use the WACCM-X model in the manuscript. We only utilized the simulation results of the WACCM-X model from Liu et al. (2023).

The detailed information of the "Tsunami simulation model" is given above.

- Lines 110-111, can you describe the variables in equations 1) and 2)?

Response:

Thank you very much for your comment. The following description is added to the revised manuscript.

where $x_i$, $k_i$, $c_{g_i}$ (i=1,2,3), and $\omega$ are the position vector, wavenumber vector, group velocities, and intrinsic frequency, respectively.

- Line 112. It is weird that "There is no real-time temperature data available in this study".

Response:

This sentence is removed from the manuscript.

- Lines 112-117, I am confused why temperature is mentioned here, which should be moved after Lines 124-125.

Response:

Thank you very much for your suggestion. Lines 112-117 are moved after Lines 124-125.

- Line 123. Why use the "HWM-14" model? Is this model suitable to study the HTHH?

Response:

Thank you for your criticism and comments. Yes, you are right. The HWM wind field is an empirical model wind field that can only reflect a long-term trend, which is not suitable for event research. Therefore, we use meteor radar wind fields to replace the HWM wind fields.

- Lines 131-149 and Figure 1. The description of Figure 1 is too general. Can you make a detailed list of the airglow (which one? Lines 133-135 only mentioned OH airglow)? The figure is very hard to follow, I have to zoom it 4 times to look carefully but it is still unclear for the wave packet in each sub-figures. Therefore it is hard to judge.

Response:

Thank you for your comments.

We have made some revisions to Figure 1 to better illustrate the wave structures. In addition, we also made a video (https://doi.org/10.5446/66190) to present the propagation process of wave structures. The airglow image shown in Figure 3 shows that, except for the 557.0 nm radiation at Lhasa station, all other stations exhibit OH airglow radiation.

[Figure]

**Figure 3** Five strong group atmospheric waves associated with the Tonga volcano eruptions were observed in the mesopause region by the ground-based airglow network. Different colored triangles correspond to each wave event sampling point, while red, blue, green, yellow, and cyan correspond to wave packet #1, #2, #3, #4, and #5, respectively. The red time markers in this figure and the following figure represent the lapse time since the volcano eruption.

- Line 158, It looks to me that the reference using the subscription 4 should not the 4ᵗʰ reference in the list.

Response:

The correct reference is Wright et al. (2022).

- Line 159, please add some references here for the surface pressure changes.

Response:

The following reference is added to the revised manuscript.

Omira, R., Ramalho, R.S., Kim, J. et al. Global Tonga tsunami explained by a fast-moving atmospheric source. Nature 609, 734–740 (2022). https://doi.org/10.1038/ s41586-022-04926-4

Takahashi, H., Figueiredo, C.A.O.B., Barros, D. et al. Ionospheric disturbances over South America related to Tonga volcanic eruption. Earth Planets Space 75, 92 (2023). https://doi.org/10.1186/s40623-023-01844-1

- Lines 159-161. It reads the logical is not clear. As mentioned Lamb wave is almost non dispersive which has purely horizontal motion. So the second sentence to Describe Figure 3 of atmospheric waves from the ionosphere to surface will cause confusion. It would be better to move or delete the first sentence.

  Response:

  The following sentence is removed from the revised manuscript.

  Figure 3 shows vertical distribution characteristics of atmospheric waves caused by Tonga volcano eruption from the surface to the thermosphere atmosphere.

- Line 189. This is not pressure profile, it is time series of surface pressure.

  Response:

  "pressure profile" is changed to "time series of surface pressure".

- Line 192, please change "position" to location.

  Response:

  "position" is changed to "location".

- Lines 204-205. "are very consistent with the simulated tsunamis near the coast"? Where have you shown the simulation result?

  Response:

  I'm sorry, we didn't express clearly. The simulation results mentioned are shown in Figure 8 in revised manuscript.

- Line 210. Can you describe Figure 5 in detail? Just saying snapshots is not enough since the contour unit is in cm.

Response:

Thank you for your comments. We give a more detailed description as follows:

Figure 8 shows snapshots of the TIAPW and TITVE simulation results. The leading TIAPW excited by the pressure disturbances travels at the same speed as the atmospheric pressure wave and is followed by subsequent sea waves generated earlier in the atmospheric pressure wave propagation which thereafter travel at the conventional tsunami propagation speed. Under a given pressure gradient, the discharge flux in deep sea is much greater than that in shallow water. A deep bathymetric feature such as the Kermadec Tonga Trench can more effectively generate tsunami waves. The wave train following the leading wave travelling over the trench appear to be larger than those travelling in other directions. The propagation speed of TITVE from the shallow water (long) wave approximation is $v = \sqrt{gH_0}$ (Salmon, 2014), where g is the gravitational acceleration and $H_0$ is the ocean depth. For sea water with a general depth of 4 km, the speed of shallow water wave is about 200 m/s. Therefore, the TIAPW is significantly faster than the TITVE. The amplitude of TITVE is greater than that of tsunamis generated by atmospheric pressure waves. The wave train following the leading wave of TITVE exhibit finer structures with scales smaller than that of TIAPW. We found that the TIAPW arrived along the coast of Chinese Mainland about 4-5 hours earlier than the TITVE.

- Lines 216-218. What the evidence (where the results, assuming Figure 5 but it is vague in its description) to support these?

Response:

Thank you very much for your comments.

Air pressure waves are not very efficient at directly exciting tsunamis in shallow water due to the weaker air-sea coupling. The Yellow sea is quite shallow, so the amplitude of the leading tsunami wave directly excited by the air wave is small there and not visible using the current colormap of the plot. The waves that you can clearly see from the air-wave generated tsunami snapshots are the subsequent waves with larger amplitudes that follow the leading wave. These subsequent waves were also generated by the air-wave but when it was traveling over the deeper ocean.

- Lines 241-243. How do you estimate the wavelength? Which one (TIAPW, TITVE) or both are consistent with the derived from the airglow network which was mentioned in Lines 204-205?

Response:

Taking TIAPW as an example, we take three lines along the wave vector direction, and each line segment takes four sampling points (red points in the Figure S2 below) at the peak position. The distance between the two adjacent sampling points is the wavelength. The average wavelength can be obtained by averaging the distances between all neighboring points

[Figure]

Figure S2

The average wavelength derived from TIAPW is 82 km ± 4km, while the average wavelength derived from the airglow network is 84 km ± 5km.

The average wavelength derived from TIAPW is 95 km ± 5km, while the average wavelength derived from the airglow network is 89 km ± 6km.

Therefore, compared to TITVE, the wavelength of TIAPW is more consistent with the results from the airglow network.

- Line 255-256. If wind comes from ERA5 (which has hourly product), why use SABER temperature?

    Response:

    Thank you for your comment. Due to the lower boundary of the SABER temperature field starting from approximately 15 km, the ERA-5 temperature field is used in areas below 15 km altitude.

- Figure 8c. Why the ray tracing calculation shows similar gradient (for example, sharp gradient around 60 km) for different sampling points A1, A2 and A3?

    Response:

Due to the almost no change in the wind field at points A1, A2, and A3, meanwhile, the temperature field comes from the same SABER temperature profile, so the trajectory changes of ray tracing are almost identical

- Line 288. "If AGWs observed by the airglow network satisfy the dispersion relation" reads weird, since the figures/main text are trying to persuade the readers the current work from the multi-layer airglow observation network has detected the information of AGWs, then lines 117-118 from the reference gives the dispersion relationship of AGWs and there is a proximate relationship in Lines 279-281…

  Response:

  I'm sorry for not expressing clearly and causing confusion to you.

  What we want to express is that the free oblique propagation characteristics of gravity waves satisfy the relationship between horizontal propagation distance and oblique propagation angle derived from dispersion relationships. We have re described as follows:

  If the AGWs observed by the airglow network propagate freely rather than being constrained by duct, we will obtain the propagation characteristics similar to that observed by Azeem et al. (2007) in the ionosphere from TEC observations.

- Line 319: SWITVE should be TTIVE?

  Response:

  "SWITVE" is changed to "TITVE".

- Line 334. Similar comment as above.

  Response:

  Thank you very much for your comment.

  This aspect has been further elaborated in the main text.

- Lines 335, re-order this sentence. Better move "in the mesopause region" after AGWs?

  Response:

  "in the mesopause region" is moved after "AGWs".

- Line 345, why "directly"? remove it.

  Response:

"directly" is removed from the revised manuscript.

- Figure 10. Can you explain more about this possible mechanism? It is too general. It is well known how important the coupling processes among ocean/land/atmosphere to study the whole atmosphere etc.

Response:

Thank you very much for your suggestion. We give a more detailed description as follows:

The 2022 HTHH volcano eruption form a complex coupling relationship in the land-ocean-atmosphere system (Fig. 14). Firstly, the heat released by the eruption has a direct impact on the ocean, causing temperature changes in the surrounding waters. This can lead to changes in the marine environment, affecting the behavior, distribution, and ecosystem structure of organisms.

Meanwhile, volcanoes release gases such as carbon dioxide and sulfur dioxide. Carbon dioxide is one of the greenhouse gases that can cause an increase in Earth's temperature, leading to global warming. Sulfur dioxide can cause sulfuric acid mist in the atmosphere, which affects the reflectivity and temperature of the atmosphere, and thus affects the global climate.

Moreover, the 2022 HTHH volcano eruptions also trigger atmospheric waves and tsunamis. The surface atmospheric pressure wave generated by the 2022 HTHH volcano eruption can affect the upper atmosphere. The conventional tsunami triggered by the Tonga volcano generated AGWs. The atmospheric pressure wave from the eruption generated a fast tsunami never before observed by tsunami observation networks. When the tsunamis reach the coast, their speeds decrease but their amplitudes increase, and the AGWs generated by them will also affect the upper atmosphere. These AGWs play an important coupling role between the ocean and the atmosphere by affecting the density and pressure distribution of the atmosphere during propagation, leading to changes in the wind field and affecting global atmospheric circulation. This study exhibits special dynamic coupling process between air and sea via acoustic gravity waves (Fig. 14). This indirect impact on the upper atmosphere provides a new perspective for us to study the coupling between the ocean and the atmosphere and a key opportunity to improve the air-sea coupling model, thereby enhancing our future ability to make tsunami warning forecasts.

[Figure]

**Figure 14** The Tonga volcano eruptions triggered two types of tsunamis, one type of tsunami is induced by the atmospheric pressure wave (TIAPW) and the other type tsunami is directly induced by the Tonga volcano eruption (TITVE). The acoustic gravity waves (AGWs) caused by tsunamis can propagate to the mesopause region.

---

## Author Comment (AC2)

Response to Referee #2

The manuscript did a very comprehensive study covering a very complex sea-air interaction through the link: volcano eruption - atmospheric pressure waves - tsunamis - gravity waves - mesosphere airglow. This manuscript provides important and likely first observations of volcano-related Lamb and gravity waves in the mesosphere airglow. In general, the authors provide evidence and arguments to support their conclusions. But I think the provided evidence is a little weak, more like a correlation between two, not robust enough to prove the generation relation. The Tonga volcano and related tsunami/waves have been well observed and reported; there should be good cross-verification between this study and previous studies. It is fine that the presented results are different from those previous studies, but you need to provide explanations. As the manuscript mentioned, many presented results are the first ever observed, so more careful reasoning and analysis are needed to support the observations. Five waves (Lamb wave L0, L1, and three gravity waves) were observed and reported in this study; I have concerns about each of them.

Thank you very much for your thoughtful, professional, and constructive comments concerning our manuscript entitled "Upper Atmosphere Responses to the 2022 Hunga Tonga-Hunga Ha'apai Volcanic Eruption via Acoustic-Gravity Waves and Air-Sea Interaction". Those comments are all valuable and very helpful for revising and improving our paper, as well as the important guiding significance to our researches. We have studied concerns carefully and have made corrections which we hope meet with approval. The detailed point-by-point responses are given below.

1. Lamb waves have a vertical phase front below the thermosphere or zero vertical wave numbers. This means what you observe in the mesosphere should look the same or very similar to what is observed in the stratosphere. Many studies have reported the Lamb wave in the brightness temperature or IR radiation from geostationary satellites such as GOES (https://agupubs.onlinelibrary.wiley.com/doi/full/10.1029/2023GL106097), Himawari-8 (https://agupubs.onlinelibrary.wiley.com/doi /full/10.1029/2022GL 098324). So, I would expect that the wave pattern in the mesosphere OH airglow looks very similar to these studies, like a solitary wave with one strong leading wave with much weaker trailing waves. What you show in the OH airglow images are stable wave trains? You show the surface pressure for the L0/L1 mode; both seem to be like solitary waves. (You should show the time derivative of the pressure to emphasize the wave.) Are you able to get the Himawari-8 brightness temperature and compare it with your OH airglow results, then you will have the full link of surface-> stratosphere->mesosphere.

Response:

Thank you very much for your comment. This is a very good question.

Yes, you are right. Lamb waves have a vertical phase front below the thermosphere or zero vertical wave numbers in the vertical direction and like a solitary wave with one strong leading wave with much weaker trailing waves in the horizontal direction, especially more pronounced in the lower atmosphere.

It is seen from model simulations that the wave amplitudes of L0 and L1 modes are not uniform at the wave front. This non-uniformity becomes more pronounced in the upper atmosphere (e.g. Figure 2 of Liu et al., 2023), probably as a result of the large variation of the background atmosphere propagation conditions. It is thus possible that over certain regions the trailing waves become comparable with the leading wave.

In addition, due to the smaller field of view of the airglow imager compared to satellite observations, some structures may be related to local fine structures, especially in the middle and upper layers where many internal waves have significant amplitudes, which may be relatively more significant than Lamb waves (Lamb waves have smaller amplitudes than internal waves).

Based on your suggestion, we have incorporated the Himawari-8 brightness temperature data (Otsuka, 2022) into this study to form a more complete observation chain of surface-> stratosphere->mesosphere (Figure 5).

From the Figure 5, it can be seen that the phase of the lamb wave is almost vertical from the ground to the stratosphere and then to the mesosphere.

[Figure]

**Figure 5** (a) OI 630 nm airglow observation at 13:13:18 UT. OH airglow network observations when (b) wave packet #1 and (c) wave packet #2 pass through the zenith direction of Xinglong Station at 13:13:34 UT and at 16:32:16 UT, respectively. (d)-(e) Himawari-8 6.2 μm brightness temperature at 13:10:00 UT. (f)The surface time series of surface pressure obtained from Xinglong observation station. The red line represents the time derivative of the pressure. The sudden change of air pressure at 13:15 UT indicates the arrival time of Lamb wave L0. A small disturbance of air pressure occurs at 16:33 UT indicates the arrival time of Lamb wave L1. The yellow stars represent the location of the Xinglong station.

References

Liu, H.-L., Wang, W., Huba, J. D., Lauritzen, P. H., & Vitt, F. (2023). Atmospheric and ionospheric responses to Hunga-Tonga volcano eruption simulated by WACCM-X. Geophysical Research Letters, 50, e2023GL103682. https://doi. org/10.1029/2023GL103682

Otsuka, S. (2022). Visualizing Lamb waves from a volcanic eruption using meteorological satellite Himawari-8. Geophysical Research Letters, 49, e2022GL098324. https://doi. org/10.1029/2022GL098324

2. Regarding the L0/L1 mode (wave #1 and #2), this manuscript seems to be one of the few that report the weaker L1 mode. You need a better estimate of the wave parameters including wavelength, period, and amplitude (airglow intensity fluctuations to be compared with L0), to justify the observations.

Response:

Thank you very much for your comment and suggestion.

Yes, you are right. The wave parameters need to be well estimated, and we use the cross spectral analysis method to estimate the wave parameters.

The detailed spectral analysis process is as follows

The airglow image was calibrated with the help of standard star map (Garcia et al., 1997) and projected into geospatial space. The background radiation is removed by differential method (Swenson and Mende,1994), to highlight atmospheric fluctuations. The atmospheric wave parameters (horizontal wavelength $\lambda_h$, observed speed $c$, and the relative intensity perturbation $I'/I$) are extracted from spectral analysis method. Figure 2c presents the two-dimensional cross spectrum obtained from Figures 2a and 2b. Zonal ($k_x$) and meridional ($k_y$) wave numbers are determined from the peak position of the spectra. The horizontal wavelengths $\lambda_h$ are obtained from the expression of $\lambda_h = 2\pi \big/ \sqrt{k_x^2 + k_y^2}$. The observed speeds $c$ are calculated from the phase ($\varphi$) (Figure 2d) at the maximum peak of the cross spectrum as $c = \dfrac{\varphi}{2\pi} \dfrac{\lambda_h}{\Delta t}$, where $\Delta t$ is the time interval between the two TD images. The amplitudes of intensity perturbations were calculated by integrating the power surrounding the central peaks of the power spectrum. To eliminate noise, the energy of the wave spectrum should be greater than 10% of the total spectrum (Tang et al., 2005).

[Figure]

**Figure 2** The time difference images (a-b) obtained from the Xinglong OH airglow imager on the night of 15 February 2022. Each image is projected on an area of 250 km × 250 km. The (c) cross spectrum and (d) phase of the time difference images from a and b using 2-D fast Fourier transform.

References

Garcia, F. J., Taylor, M. J., and Kelley, M. C.: Two-dimensional spectral analysis of mesospheric airglow image data, Appl. Optics, 36 (29), 7374–7385, 1997.

Swenson, G. R. and Mende, S. B.: OH emission and gravity waves (including a breaking wave) in all-sky imagery from Bear Lake, UT, Geophys. Res. Lett., 21, 2239–2242, 1994.

Tang, J., Kamalabadi, F., Franke, S. J., Liu, A. Z., and Swenson, G. R.: Estimation of gravity wave momentum flux with spectroscopic imaging, IEEE T. Geosci. Remote, 43, 103–109, 2005.

3. Also, is there any evidence for the ~3-hour separation between L0 and L1? Wright et al. 2022 report what they called primary and secondary Lamb waves (not sure if they match L0/L1), but the time difference is about 60-min. This study (https://agupubs.onlinelibrary.wiley.com /doi/10.1029/2023 GL103809) presents L1 mode also in the mesosphere. From wind measurements, the Lamb wave L1 mode period

(2-hr), wavelength (1400 km), and wave pattern (a solitary wave) seem not to match what is presented in the airglow. And very interesting, they do not see L0 mode and argue that L0 mode is likely a higher-frequency wave and got averaged out.

Response:

Thank you very much for your comment. This is a good question. We think that the time interval between L0 and L1 is not a constant, but rather anisotropic due to the influence of the background atmosphere. Thank you very much for providing the references Sepúlveda et al. (2023) and Poblet et al. (2023).

According to Figure 1 of Sepúlveda et al. (2023). We can estimate that the time it takes for the L0 to reach South America is approximately 13 UT, on January 15, 2022, while the time it takes for L1 to reach South America is approximately 18 UT (Poblet et al., 2023), with a time interval of approximately 5 hours between the two wave modes.

[Figure]

Figure 1 of Sepúlveda et al. (2023)

[Figure]

Figure 2 of Poblet et al. (2023)

In the northern hemisphere, over Chinese Mainland, Li et al. (2023) used dense global navigation satellite system data from China to track the propagation of traveling ionospheric disturbances (TIDs) triggered by the 2022 January 15 Tonga volcanic eruption.

They found two group of TIDs corresponding to L0 and L1, respectively. The time for TID related to L0 to reach the Chinese Mainland is about 11:00 UT, while the time for TID related to L1 to reach the Chinese Mainland is about 13:30 UT. The time interval between the two is 2.5 hours, which is very close to the 3-hour interval we observed.

References

Li, X., Ding, F., Yue, X., Mao, T., Xiong, B., & Song, Q. (2023). Multiwave structure of traveling ionospheric disturbances excited by the Tonga volcanic eruptions observed by a dense GNSS network in China. Space Weather, 21, e2022SW003210. https://doi. org/10.1029/2022SW003210

Poblet, F. L., Chau, J. L., Conte, J. F., Vierinen, J., Suclupe, J., Liu, A., & Rodriguez, R. R. (2023). Extreme horizontal wind perturbations in the mesosphere and lower thermosphere over South America associated with the 2022 Hunga eruption. Geophysical Research Letters, 50, e2023GL103809. https://doi. org/10.1029/2023GL103809

Sepúlveda, I., Carvajal, M., & Agnew, D. C. (2023). Global winds shape planetary-scale Lamb waves. Geophysical Research Letters, 50, e2023GL106097. https://doi.org/10.1029/2023GL106097

4. For tsunami-generated gravity waves (#3—#5 in this manuscript), they have been studied in many model simulations. (https://agupubs.onlinelibrary.wiley.com/doi/full/10.1029/2022JA030301, https://agupubs. onlinelibrary.wiley.com/doi/10.1029/2020JA028309, https://agupubs.onlinelibrary.wiley.com/doi/full/10.1002/2016JD025 673). However, it seems the ocean surface vertical displacements need to be large enough (10 cm or even higher) to produce 1% airglow fluctuations, Table 3 of Inchin et al. 2022). Otherwise, the wave signature might not be detectable in the mesosphere airglow. This is another reason that you need to carefully estimate the observed wave amplitudes in airglow fluctuation percentage. Another drawback of this part is the tsunami simulations were not verified, such as by DART buoy wave height measurements as well as many published studies.

I am very skeptical about the gravity waves being tsunamis generated if the surface wave height is in the order of 2 cm, as shown in Figure 6. Weak tsunamis likely won't generate atmospheric gravity waves. It might even be the opposite; those weak tsunamis are possibly excited by atmospheric gravity waves. (https://www.nature.com/articles/s41586-022-04926-4 and https://www.nature.com/articles/s41598-022-25854-3). This Figure (https://www.nature.com/articles/s41586-022-04926-4/figures/6) demonstrates the observed tsunamis are very well simulated. The order of magnitudes of the tsunami's surface height is somewhat similar to the 2 cm for two sites, 21416 and 23219, both similar distances to the East China Sea from the Tonga volcano.

Response:

Thank you very much for your comment. This is a great question. There have been theoretical (Peltier and Hines, 1976) and observational (Grave and Makela, 2015, 2017) studies on the relationship between the amplitude of tsunamis and gravity waves.

Peltier and Hines (1976) found that a tsunami amplitude of ± 1 cm at sea level can cause vertical motion of ionospheric E layer and F layer ± 100 m. A more direct observational evidence is that Grawe and Makela (2017) provided airglow observation of tsunami-generated ionospheric signatures over Hawaii caused by the 16 September 2015 Illapel earthquake. They found that vertical disturbances on the sea surface not exceeding 2 cm (Figure 3b of Grave and Makela (2017)) can create detectable signatures in the ionosphere (Figure 1 of Grave and Makela (2017)).

Therefore, there is no doubt that the generation of gravity waves by this sea surface variation (the order of 2 cm) and their propagation to the middle atmosphere are credible.

The models for the 2022 Hunga Tonga-Hunga Ha'apai Volcanic Eruption used in our study was estimated and validated with observations at offshore DART stations around the Pacific Ocean in a previous study (Figure 3 and Figure 7 of Gusman et al., 2022; https://link.springer.com/article/10.1007/s00024-022-03154-1).

On the other hand, we have also demonstrated through ray tracing method that the waves observed in the airglow originate from simulated tsunami waves. So we believe that our tsunami simulation results are reliable.

[Figure]

**Figure 1.** All frames of the filtered 630.0 nm airglow with a visible signature during the arrival of the tsunami to Hawaii on 17 September 2015. The structure is propagating to the northwest. The red diamond in the top left image shows the location of DART station 51407.

Figure 1 of Grawe and Makela (2017)

[Figure]

Figure 3b of Grawe and Makela (2017)

References

Grawe, M. A., and J. J. Makela (2015), The ionospheric responses to the 2011 Tohoku, 2012 Haida Gwaii, and 2010 Chile tsunamis: Effects of tsunami orientation and observation geometry, Earth and Space Science, 2, 472–483, doi:10.1002/2015EA000132.

Grawe, M. A., and J. J. Makela (2017), Observation of tsunami-generated ionospheric signatures over Hawaii caused by the 16 September 2015 Illapel earthquake, J. Geophys. Res. Space Physics, 122, 1128–1136, doi:10.1002/2016JA023228.

Peltier, W. R., and C. O. Hines (1976), On the possible detection of tsunamis by a monitoring of the ionosphere, J. Geophys. Res., 81(12)

5. Many other studies that report the Tonga eruption-related waves did a statistical analysis to verify the observed perturbations on that day were not random or just background (they exist even if there is no volcano eruption). Basically, they take a long-term average of several days to get a background perturbation. In your case, make sure you do not see those similar waves every day. If you observe something different before and after that day, then you can tell what you observe on that day is eruption-related.

For the reason above, the selected snapshot-style airglow images at several time steps are not very robust evidence for such volcano/tsunami-related waves. Keograms or movies can better demonstrate the arrival of the waves. Also, you need to add the distance to the volcano on the map.

Response:

Thank you very much for your suggestion. This is a very good question.

According to your suggestion, we have made animations of the day of the Tanga volcanic eruption(https://doi.org/10.5446/66190), as well as two days before and after Tanga eruption (https://av.tib.eu/series/1689; Movie S1 corresponds to January 13th, Movie S2 corresponds to January 14th, Movie S3 corresponds to January 16th, and Movie S4 corresponds to January 17th), as supporting materials. Through animation, it can be found that the observation of the airglow is very calm before and after Tanga eruption, while on the day of the eruption, severe disturbances were observed and the wave propagation direction comes from the Tonga direction. Therefore, there is a strong correlation between the atmospheric fluctuations observed on the day of the Tanga eruption and the Tonga volcanic eruption.

At the same time, we have also added the following Figure, including the station name and distance from the Tonga volcano.

[Figure]

**Figure 1** The distribution of airglow network stations, along with the large circular centered on the Tonga volcano and its radius length, is also marked in the figure

Section 3.2, Figures 5 and 6 present results about the tsunami simulation, but not many details are included, are they verified?

Response:

Thank you very much for your comment. This is a very good question.

It is necessary to verify the models through observation results. As mention above, the models for the 2022 Hunga Tonga-Hunga Ha'apai Volcanic Eruption used in our study was estimated and validated with observations at offshore DART stations around the Pacific Ocean in a previous study (Figure 3 and Figure 7 of Gusman et al., 2022; https://link.springer.com/article/10.1007/s00024-022-03154-1).

On the other hand, we have also demonstrated through ray tracing method that the waves observed in the airglow originate from simulated tsunami waves in the coastal areas of China. So we believe that our tsunami simulation results are reliable.

All figures and corresponding discussions: Add the lapse time since the volcano eruption in all the figures, which could better help to understand the evolution of the wave pattern.

Response:

This is a very good suggestion. We add the lapse time since the volcano eruption in the figures.

21, airglow imaging system network or airglow imager network.

Response:

"a ground-based airglow network" is changed to "a ground-based airglow imager network".

22, I anticipate this 'phase speed' is horizontal phase speed or total phase speed.

Response:

Yes, you are right. "phase speed" is changed to "horizontal phase speed".

24, confusing, L0 and L1 wavefront is vertical, then L0 wavefront mode tilts forward.

Response:

The characteristics of the theoretically (Lindzen and Blake, 1972) predicted L0 mode is that the phase lines of the wavefront are vertical up to the lower thermosphere and tilt outward above. As for Lamb wave L1 mode, the ground and mesopause region provide waveguide surfaces, resulting in maximum wave energy between the two layer, while the phase does not change with height (Francis, 1973).

References

Francis, S. H.: Acoustic-gravity modes and large-scale traveling ionospheric disturbances of a realistic, dissipative atmosphere, J. Geophys. Res., 78 (13), 2278– 2301,1973.

Lindzen, R. S., and Blake, D.: Lamb waves in the presence of realistic distributions of temperature and dissipation, Journal of Geophysical Research, 77(12), 2166–2176, 1972.

42-44, if you explicitly mention 'this volcanic eruption,' I would expect that those waves are all reported from this eruption, so what is the purpose of those old references, even if they are very classic?

Response:

Thank you for your comments. Those old references are removed from the manuscript.

54, GNSS TEC, better to give the full name, at least for TEC.

Response:

Thank you very much for your suggestion.

The full name of GNSS TEC is global navigation satellite system (GNSS)- total electron content (TEC), which is added to the manuscript.

73, conventional tsunami, sounds unclear here.

Response:

Thank you for your comments.
We have defined "conventional tsunami" as follows
Conventional tsunamis are typically generated by localized sea surface displacements caused by sources such as earthquakes and volcanoes, similar to the tsunamis directly induced by the 2022 Tonga volcano eruption (TITVE).

74, only two studies, maybe because they are rare.

Response:

"only two studies" is changed to "only two rare studies".

83, better use one extra sentence to describe the term in ().

Response:

Thank you very much for your suggestion. "(air-water-air-coupling process)" is changed to "through air-water-air-coupling process".

90-91, I guess there are two filters used on the same lens; basically, it is the same equipment. It is a little confusing to declare two airglow networks. Also, clearly state what types of airglow are observed.

Response:

Thank you very much for your comments.

In order to achieve higher temporal resolution, they are independent of each other, meaning that one filter corresponds to one lens. So, various types of airglow, including OH, OI 557.0 nm, 630.0 nm, and Na airglow have been observed. But in this study, we mainly used OH and OI 630.0 nm airglow.

105, what is "moving change pressure"

Response:

The moving change pressure is used as input for the momentum equation of tsunami simulation. It changes with distance and time from the center of the Tonga volcanic eruption (Gusman et al., 2022).

References

Gusman, A. R., Roger, J., Noble, C. et al. The 2022 Hunga Tonga-Hunga Ha'apai Volcano Air-Wave Generated Tsunami, Pure and Applied Geophysics, 2022.

107, I would expect to see a brief here about the tsunami simulation and some more details like section 2.3.

Response:

Thank you very much for your comments. We have provided a more detailed description of the tsunami model simulations

2.2 Tsunami simulation model

Tonga submarine volcano erupted on 15 January 2022, and generated tsunamis that were detected around the globe, affected particularly the Pacific region. In this study, two types of tsunamis were simulated, conventional tsunami simulations and atmospheric pressure wave-induced tsunami simulations. The linear-shallow water equations in the spherical coordinate system are used to simulate the tsunamis from the localized source and atmospheric pressure wave. The continuity equation of a linear shallow water wave model in spherical coordinates is:

$$\frac{\partial \eta}{\partial t} + \frac{1}{R\sin\theta}\left[\frac{\partial(ud)}{\partial\varphi} + \sin\theta\frac{\partial(vd)}{\partial\theta}\right] = 0 \tag{1}$$

where $\eta$ is free surface elevation (m), $d$ is the water depth (m), $R$ is the Earth's radius (6371,000 m), $\varphi$ is longitude, $\theta$ is colatitude.
While the momentum equations of the linear shallow water wave model are:

$$\frac{\partial u}{\partial t} + \frac{1}{R\sin\theta}\left[g\frac{\partial\eta}{\partial\varphi} + \frac{1}{\rho}\frac{\partial p}{\partial\varphi}\right] + fv = 0 \tag{2}$$

$$\frac{\partial v}{\partial t} + \frac{1}{R}\left[g\frac{\partial\eta}{\partial\theta} + \frac{1}{\rho}\frac{\partial p}{\partial\theta}\right] - fu = 0 \tag{3}$$

where, $u$ is the velocity along the lines of longitude (m/s), $v$ is the velocity along

the lines of latitude, $g$ is the gravitational acceleration (9.81 m/s$^2$ ), $p$ is the atmospheric pressure (Pa), $\rho$ is the sea water density (1026 kg/m3 ), $f$ is the Coriolis coefficient. For the atmospheric pressure wave-induced tsunami simulation, the moving change pressure terms as an input to tsunami simulation momentum equation. The atmospheric pressure wave model is based on the Equation (1) in Gusman et al. (2022).

For the tsunami simulations from a localized source, a B-spline function (Koketsu and Higashi, 1992) below is used to represent the circular water uplift source at the volcano:

$$f(x, y) = \sum_{i=0}^{3} \sum_{j=0}^{3} c_{k+i, l+j} B_{4-i}(\frac{x - x_k}{h}) B_{4-j}(\frac{y - y_l}{h}) \tag{4}$$

where

$$B_i(r) = \begin{cases} r^3/6, & i = 1 \\ (-3r^3 + 3r^2 + 3r + 1)/6, & i = 2 \\ (3r^3 - 6r^2 + 4)/6, & i = 3 \\ (-r^3 + 3r^2 - 3r + 1)/6, & i = 4 \end{cases} \tag{5}$$

$x_k$ and $x_l$ stand for the coordinates of the knots along the x and y axes, h is the characteristic diameter of water uplift, $r$ is the great-circle distance from the volcano eruption center, $c_{1,1} = 1$ and the other $c_{k+i, l+j} = 0$. In this study, the modelling domain covers the Pacific Ocean and some parts of Indian Ocean and the Caribbean with a grid size of 5 arc-min. For detailed tsunami simulation algorithms, please refer to Gusman et al. (2022).

125, is there any detrending or filtering on the SABER temperature before it is used in the calculation?

Response:

The temperature product provided by TIMED/SABER team directly represents the background atmospheric temperature and does not require detrending or filtering processing.

Line 141, do not use the alphabet o for a degree; use the math symbol.

Response:

It has been modified.

Section 3.1 and Figure 1, better give each station a name or a number, so you could clearly state the wave front was visible in which station.

Response:

Thank you very much for your good suggestion.
We have added the following Figure 1, including the station name and distance from the Tonga volcano.

[Figure]

**Figure 1** The distribution of airglow network stations, along with the large circular centered on the Tonga volcano and its radius length, is also marked in the figure

Figure 1, in sub-Figure 6 (middle row, right), P1-P6 are labeled; what are they? In sub-Figures 8 and 9, what are early and late wave packets?

Response:

We have made adjustments to Figure 1 (Figure 3 in revised manusrcipt) and provided explanations for the markings in it.

[Figure]

**Figure 3** Five strong group atmospheric waves associated with the Tonga volcano eruptions were observed in the mesopause region by the ground-based airglow network. Different colored triangles correspond to each wave event sampling point, while red, blue, green, yellow, and cyan correspond to wave packet #1, #2, #3, #4, and #5, respectively. The red time markers in this figure and the following figure represent the lapse time since the volcano eruption.

Figure 1: the labels are often obscured by the airglow images on the map, hard to read. There are plenty of blank areas between airglow images where you can put those labels.

Response:

Thank you very much for your suggestion.

We have put the labels in blank areas.

Figure 1: what happened in sub-Figure 1-6 (top and middle rows) for the station over the 110E and 20S, on that island?

Response:

Thank you very much for your carefully review.

We have reprocessed the data over the 110E and 20S.

Figure 1: for the station over the 120E, 25S, it seems it does not provide much information due to weather.

Response:

Thank you very much for your comment.

Yes, you are right. This is due to weather conditions. To maintain the completeness of the data, we added them to the Figure.

Figure 1, after I checked all the information on the figure, I went to find if any supplementary materials, such as videos showing the motion of the 5 wave packets, are available. Unfortunately, I did not find any. Can the authors make some videos from the airglow images?

Response:

Thank you for your criticism and comments
According to your suggestion, we have made videos of the day of the Tanga volcanic eruption(https://doi.org/10.5446/66190), as well as two days before and after Tanga eruption (https://av.tib.eu/series/1689), as supporting materials.

149-155, how are those wave phase speeds and amplitudes estimated? For reference, what is the speed of sound at the same altitudes?

Response:

As mention above, the wave parameters are obtained from cross spectral analysis. As your suggestion, we calculate the speed of sound using the following equation

$$c_s = \sqrt{\gamma R T} \quad \gamma = 1.4 \quad R = 287.06 \text{ J/(kg·K)}$$

We found that the sound speed at the height of the OH airglow is approximately 294 m/s, slightly lower than the speed of lamb wave L0 mode at this height.

Figure 2: you need to clarify the P1-P6 points, even if you show them in Figure 1. Why only information of wave #1-#3 is shown, not wave #4-#5.

Response:

Thank you very much for your comment.

The original Figure 2 has been replaced by the following figure.

Figure 4 shows the distribution of wave parameters for multi-group of atmospheric waves (wave packet #1-#5). The phase speed of wave packet #1 is approximately 309 m/s. Wave packet #2 displays a slightly slower phase speed, with average phase speed of 236 m/s. The horizontal phase velocity of group wave packet # 3-5 is less than that of the first GW, which is mainly distributed in the range of 200 m/s to 215 m/s. The horizontal wavelengths of these five group wave packets are mainly distributed in 80 km-105 km, while the observation period is relatively small and mainly concentrated in 5.7 min-7.2min. For amplitude, the average amplitude of the lamb wave L1 mode (5.4%) is higher than that of the lamb wave L0 mode (3.2%). Wavepacket # 3, # 4, and # 5 have relatively small amplitudes, mainly distributed between 0.85% and 1.25%.

[Figure]

**Figure 4** Distribution of (a) wave wavelength, (b) velocity, (c) period, and (d) amplitude parameters for multi-group of atmospheric waves (wave packet #1-#5). The calculation of wave packet parameters comes from the average value of the wave passing through the sampling points in Fig 3.

Figure 2: wave amplitudes in the airglow intensity need a better qualification using the percentage of the airglow intensity fluctuations rather than arbitrary units.

Response:

Thank you very much for your suggestion.

The original Figure 2 has been replaced by the following figure. The amplitude is represented as a percentage instead of the original arbitrary units.

159-161, this statement about vertical distribution is too much for what you showed in Figure 3, where only wave information from three layers is shown, and only Lamb waves.

Response:

Thank you for your comments

The following sentence may cause confusion and has been removed from the revised manuscript.

Figure 3 shows vertical distribution characteristics of atmospheric waves caused by Tonga volcano eruption from the surface to the thermosphere atmosphere.

167, are you able to verify the results by estimating the speed * travel time to be the distance between two sites?

Response:

Thank you very much. This is a very good suggestion.

Figure S1 (below) presents that the time it takes for Lamb L0 mode to reach the zenith direction of Zhangzhou station is approximately 12:19:57 UT, and the time it takes to reach the zenith direction of Xinglong station is approximately 13:13:34 UT, with a time interval of approximately 53.5 min. From Figure 1, it can be seen that the radial distance between the two stations with Tonga volcano as the center is approximately 9700 km. Therefore, we can estimate the velocity of a to be approximately 304 m/s, which is very close to the results obtained from cross spectral analysis.

[Figure]

[Figure]

**Figure S1**

179-185, the scale height between two types of waves only accounts for how fast the wave energy attenuates with respect to altitude; you need wave amplitudes at the source level to compare the observed wave amplitudes at the mesosphere.

Response:

Thank you very much for your good suggestion.

Yes, you are right. We need wave amplitudes at the source level to compare the observed wave amplitudes at the mesosphere.

As your suggestion, we performed derivative processing on the pressure data, as shown by the red line in Fig 5f. We found that the pressure disturbances corresponding to lamb0 and lamb1 are P1_surface=0.95 hPa and P2_surface=0.19 hPa, respectively. Assuming the scale height H=8 km, we estimated the amplification of the two reaching the OH airglow layer with $e^{\kappa z/H}$ and $e^{z/2H}$ as growth rates. $\kappa = (\gamma - 1)/\gamma$, and $\gamma$ is the ratio of specific heats (~1.4). The amplitude of internal waves increases with height at a rate greater than that of Lamb wave L0 mode. Through calculation, we found that

P1_OH= P1_surface* $e^{\kappa z/H}$ =21.28 hPa, P2_OH= P2_surface * $e^{z/2H}$ =43.83 hPa.

Although the pressure corresponding to Lamb wave L1 mode on the surface is much lower than that corresponding to L0 mode, at the height of the OH airglow layer, the pressure corresponding to L1 mode is twice that of L0 mode.

[Figure]

**Figure 5f**  The surface time series of surface pressure obtained from Xinglong observation station. The red line represents the time derivative of the pressure. The sudden change of air pressure at 13:15 UT indicates the arrival time of Lamb wave L0. A small disturbance of air pressure occurs at 16:33 UT indicates the arrival time of Lamb wave L1.

Figure 3: the band in OI airglow image is very similar to the overexposure due to clouds or reflections. Still, I would like to see some supporting evidence, like a Keogram or a video. It is better to use the derivative of the pressure to isolate the waves.

Response:

Thank you for your comments.
We carefully verified the data again and created a continuous image below (Figure S2) and an animation(https://doi.org/10.5446/66280). We are confident that it is a wave rather than a cloud.

[Figure]

**Figure S2**

224, many blank spaces are missing; better proofreading is needed.

Response:

Thank you for your comments.

We have carefully checked the entire text to ensure that there are no missing spaces.

Figure 4: yellow lines mark other wavefronts, which is the wave #3.

Response:

I'm sorry we didn't describe clearly. The yellow lines mark the wave #3.

[Figure]

**Figure 6** The red solid lines indicate leading wave front of the wave packet #2. The yellow solid lines mark wave packet #3, which are clearly not parallel to the wave fronts of wave packet #2.

Figure 5: up to 5 cm wave height for both types of tsunamis, see comments at the beginning.

Response:

Thank you very much for your comment.

As discussed above, there have been theoretical and observational studies on the relationship between the amplitude of tsunamis and gravity waves. They found that vertical disturbances on the sea surface not exceeding 2 cm can create detectable signatures in the ionosphere. The amplitude of vertical sea level motion in our tsunami simulation is an order of 2 cm (In order to highlight the sea wave, we have limited the display scale to ± 2cm, the actual amplitude is greater than 2 cm). Therefore, there is no doubt that the generation of gravity waves by this sea surface variation and their propagation to the middle atmosphere are credible.

In terms of models validation, the models for the 2022 Hunga Tonga-Hunga Ha'apai Volcanic Eruption used in our study was estimated and validated with observations at offshore DART stations around the Pacific Ocean in a previous study (Figure 3 and Figure 7 of Gusman et al., 2022; https://link.springer.com/article/10.1007/s00024-022-03154-1).

249, how do you calculate the $m^2$, which formula was used?

Response:

The calculation of the m$^2$ is from the Equation below:

$$m^2 = \frac{\omega^2}{c_s^2}(1 - \frac{\omega_a^2}{\omega^2}) - k^2(1 - \frac{\omega_b^2}{\omega^2})$$

where k is the horizontal wave number from airglow observation, $c_s$ the local speed of sound, $\omega = k(c-u)$ is intrinsic frequency, c is the horizontal phase speed, u is the background wind speed in the direction of wave propagation from meteor radar observations and ERA-5. $\omega_a^2 = \frac{g}{T}\frac{dT}{dz} + \frac{\gamma g}{4H}$ is acoustic cutoff frequency,

$\omega_b^2 = \frac{g}{T}\frac{dT}{dz} + \frac{(\gamma-1)g}{\gamma H}$ is buoyancy frequency, g is the gravitational acceleration, and T

is temperature from the Sounding of the Atmosphere using Broad band Emission Radiometry (SABER) instrument on the Thermosphere Ionosphere Mesosphere Energetics and Dynamics (TIMED) satellite.

The following References that you recommended has been added to the manuscript.

Inchin, P. A., Heale, C. J., Snively, J. B., and Zettergren, M. D.: The dynamics of nonlinear atmospheric acoustic-gravity waves generated by tsunamis over realistic bathymetry, Journal of Geophysical Research: Space Physics,125, 2020.

Inchin, P. A., Heale, C. J., Snively, J. B., and Zettergren, M.D.: Numerical modeling of tsunami-generated acoustic-gravity waves in mesopause airglow, Journal of Geophysical Research: Space Physics,127, 2022.

Laughman, B., D. C. Fritts, and T. S. Lund (2017), Tsunami-driven gravity waves in the presence of vertically varying background and tidal wind structures, J. Geophys. Res. Atmos., 122, 5076-5096, doi:10.1002/2016JD025673.

Nishikawa, Y., Yamamoto, My., Nakajima, K. et al. Observation and simulation of atmospheric gravity waves exciting subsequent tsunami along the coastline of Japan after Tonga explosion event. Sci Rep 12, 22354, 2022. https://doi.org/10.1038/s41598-022-25854-3

Omira, R., Ramalho, R.S., Kim, J. et al. Global Tonga tsunami explained by a fast-moving atmospheric source, Nature 609, 734–740, 2022. https://doi.org/10.1038/s 41586-022 - 04926-4

Otsuka, S. (2022). Visualizing Lamb waves from a volcanic eruption using meteorological satellite Himawari-8. Geophysical Research Letters, 49, e2022GL098324. https://doi. org/10.1029/2022GL098324

Poblet, F. L., Chau, J. L., Conte, J. F., Vierinen, J., Suclupe, J., Liu, A., & Rodriguez, R. R. (2023). Extreme horizontal wind perturbations in the mesosphere and lower thermosphere over South America associated with the 2022 Hunga eruption. Geophysical Research Letters, 50, e2023GL103809. https://doi.org/10.1029/2023GL103809

Sepúlveda, I., Carvajal, M., & Agnew, D. C. (2023). Global winds shape planetary-scale Lamb waves. Geophysical Research Letters, 50, e2023GL106097. https://doi.org/10.1029/2023GL106097

Wright, C. J., et al. Surface-to-space atmospheric waves from Hunga Tonga-Hunga Ha'apai eruption, Nature, 609 (7928), 741–746, 2022.

---

## Author Response (AR2)

**Response to Referee #2**

The authors carefully addressed the comments from the reviewers and made substantial changes to the manuscript. I appreciate that the authors spent time creating the movies. They are very helpful in interpreting the waves. What is the exposure time or temporal resolution? Section 2.1 states that it is 1 minute. However, the movie presented in the supplementary materials shows it is roughly 3 or 4 minutes from the time stamp on the movie. Did some images get skipped? This is important because it is related to the determined wave period of 6 minutes. Please confirm the time resolution.

Response:

Thank you very much for your careful comment.
Yes, you are right. We skipped some images because Lhasa station (29.7°N, 91.0°E) did not have OH airglow observation. We used OI 557 nm airglow observation (3 min time resolution) as a substitute in the video. In order to maintain a consistent rhythm in the animation, the animation time resolution was adopted at 3 min. Nevertheless, we have created a new video with high temporal resolution (1 min time resolution) (http://doi.org/10.5446/67795), please check.

What is the yellow box in Figure 2? Did you choose this area to apply the 2D FFT to get the wavenumber and phase?

Response:

Thank you very much for your careful comment.
Yes, the yellow box areas are used to obtain the wavenumber and phase spectrum using 2D FFT.

(Please check the caption of Figure 2 in revised manuscript with track)

To compare Figure 5(e) one-to-one, it is better to zoom it into the same map area as Figure (a-c).

Response:

Thank you very much for your suggestion.
We have made modifications to Figure 5e, please check.

[Figure]

Figure 5

Checking the airglow movies and looking at the coherence and spatial coverage of the wave pattern among continuous images, I am convinced that wave packets 3 to 5 should be related to some larger-scale wave sources toward SE, the oceanic area. If further ray-tracing simulation can prove they are tracked to be related to a tsunami by independent simulation, then it is fair to say those waves are tsunami-generated.

Response:

Thank you for your keen insight and unique perspective, which will be of great help to our research. We carefully checked the wave packet events, as shown in Figure S1 below. The appearance times of wave packets # 3, # 4, and # 5 are independent of each other. The time interval between the appearance of wave packets # 3 is 15:54 UT-16:14 UT, with a shorter duration of only about 20 minutes. And wave packets # 4 appeared in the northern network 4 hours later. And wave packets # 5 appear in the southern and

western network. Therefore, the sources of these three groups of waves are also independent and separate. However, as we discussed in the manuscript, large tsunami waves are split into smaller scale tsunamis after passing through some islands along the southeastern coast of China. Therefore, it can also be considered that the sources of these waves come from the small-scale waves decomposed from large-scale waves.

[Figure]

Figure S1

Back to wave packets #1 and #2 about Lamb wave mode L0 and mode L1. It seems this is still an open question about whether Lamb waves can reach the mesosphere and be detected there.

https://acp.copernicus.org/articles/24/4851/2024/acp-24-4851-2024.html
https://angeo.copernicus.org/articles/41/197/2023/angeo-41-197-2023.html

These two papers, from the same author, used Meteor radar data to analyze the wave signature in the MLT region. It seems they could not confirm that the Lamb wave reached the upper atmosphere. The following is a quote from the paper:

Based on our observations, we can almost rule out that the primary lamb wave that was caused by the volcanic eruption reached the upper atmosphere. Thus, the thermospheric/ionospheric observations are likely the result of multistep vertical coupling processes as described in Becker and Vadas (2018), Vadas and Becker (2018), and Vadas et al. (2018). However, the wind measurements indicated several other signals exhibiting a similar morphology, and the lifetimes of other peaks in the winds could not be linked to the lamb wave excited by the Hunga Tonga–Hunga Ha′apai eruption.

https://agupubs.onlinelibrary.wiley.com/doi/full/10.1029/2023GL103809

However, this paper supports the L1 Lamb wave signature in the meteor radar, aka mesosphere.

Response:

Thank you very much for providing a new theoretical perspective, which is very meaningful for improving our work.

Yes, you are right, Stober et al. (2018, 2024) found through meteor radar wind field observations that the anomalous peak signal cannot be solely determined to be caused by the Lamb wave generated on the Tonga volcanic eruption. In other words, it is debatable whether the Lamb wave can directly propagate to the upper atmosphere as the only way.

Therefore, reaching the upper atmosphere through secondary waves or high-order of multistep vertical coupling process is also a possible way

As for lamb wave L1 mode, as discussed in the manuscript, the lamb wave L1 mode can be regarded as internal gravity waves.

(Please check lines 263-291; 473-478 in revised manuscript with track)

It is well known that the airglow imaging system is more likely to capture high-frequency waves, especially if you use the time-difference method. The meteor radar would average out those high-frequency waves.

So the key question here: Do we believe the airglow imagers captured higher frequency waves are lamb waves?

Response:

Thank you very much for your comment.

As mentioned above. Our OH airglow observation has a time resolution of 1 minute. Therefore, our airglow observation has the ability to observe high-frequency waves. If high-frequency lamb waves can reach the upper atmosphere, the airglow imagers are able to capture this high-frequency fluctuation.

Of course, we cannot rule out that the high-frequency fluctuations observed by our OH airglow imager instrument are caused by the dissipation of the leading waves or secondary lamb waves generated by the primary lamb waves in the lower atmosphere.

You have estimated the following wave parameters: 300 m/s phase speed, 6-min period,

100 km wavelength.

The 300 m/s wave speed seems to be good in the acoustic range. The 6-minute period is also good and might explain why they are not captured by the meteor radar. My main concern is on the 100-km wavelength. If we carefully check Liu et al. 2023, the WACCM-X simulation indeed presented continuous wave trains above 31 km and a solitary wave pattern below. However, the wavelength does not vary much between solitary waves and wave trains. For wave packet #1 presented in this study, the wavelength estimated from the Himawari-8 satellite brightness temperature should be much larger than 100 km, as well as the OI airglow at about 250 km altitude. What kind of background condition between the stratosphere and mesosphere (30-80 km) would lead to waves of several hundred km wavelengths "breaking" into a much smaller scale?

Response:

Thank you very much for providing a unique perspective. We believe that wave trains may come from two mechanisms, one of which is the energy leakage of the solitary waves. Solitary waves dissipate energy by generating wave trains; Large scale main wave breaking in the stratosphere generates small-scale secondary waves that propagated to the upper atmosphere.

We observed wave trains following the leading wave front with a horizontal wavelength of ~ 120 km from Himawari-8 satellite brightness temperature observation (the area pointed by the yellow arrows in Figure S2). We also observed waves with a horizontal wavelength of ~ 400 km from Himawari-8 satellite brightness temperature observation (the area pointed by the red arrows in Figure S2).

Besides, as you mentioned, the wave features are mostly captured by stations near 40N and 120E. So, it is difficult to capture large-scale waves like satellite observations (much larger than 100 km).

[Figure]

Figure S2

From Figure 10, the winds between 40-80 km are mostly eastward and southward, so the Lamb waves propagate against the background winds. Could the wind explain something about the wavelength?

Response:

Thank you very much for your careful comment.

According to the theory of wind field filtering for atmospheric waves, waves propagating downstream may experience wind field filtering effects and dissipate into the background atmosphere, while atmospheric waves propagating upstream are easier to propagate, with increased vertical wavelengths and easier to be observed by airglow imager instruments, and have a smaller impact on horizontal wavelengths.

Wave #2 (Lamb L1) comes together with the gravity waves; it looks very much like gravity waves, except for the slight phase front orientation differences.

Response:

Thank you very much for your careful comment.

As discussed in the manuscript, the lamb wave L1 mode can be regarded as internal gravity waves.

(Please check lines 275-277, 473-478 in revised manuscript with track)

Also, it is disappointing that the wave #1 and #2 features are mostly captured by stations near 40N and 120E, making them appear like localized features. The wave signatures are very vague in the two coastal stations, so it is hard to tell the actual scale of the waves.

Response:

Yes, optical observations are easily affected by weather conditions. Nevertheless, clear waves (Figure S3) can still be seen in the observation gaps of observation stations with poor weather conditions.

[Figure]

Figure S3

In summary, I am still unconvinced that waves #1/#2 are Lamb waves, but I could not provide clear arguments for why they are not. If this paper were purely about gravity waves #3-#5, it would be solid and comprehensive. I do not want to stop the manuscript from being published, but I would recommend the authors down-tune the conclusive remarks about wave #1/#2 since they are still debatable.

Response:

Thank you very much for your comments and suggestions. It is of great significance to comprehensively improve this manuscript. Your meticulous research spirit will also be of great benefit to our future research. The question of whether lambs can propagate

directly to the upper atmosphere is an open question and cannot provide a unique conclusion. We have conducted in-depth and extensive discussions on possible propagation mechanisms to make the manuscript more comprehensive and error free.

(Please check lines 263-291; 473-478 in revised manuscript with track)

The following references has been added to the manuscript.

Stober, G., Vadas, S. L., Becker, E., Liu, A., Kozlovsky, A., Janches, D., Qiao, Z., Krochin, W., Shi, G., Yi, W., Zeng, J., Brown, P., Vida, D., Hindley, N., Jacobi, C., Murphy, D., Buriti, R., Andrioli, V., Batista, P., Marino, J., Palo, S., Thorsen, D., Tsutsumi, M., Gulbrandsen, N., Nozawa, S., Lester, M., Baumgarten, K., Kero, J., Belova, E., Mitchell, N., Moffat-Griffin, T., and Li, N.: Gravity waves generated by the Hunga Tonga–Hunga Ha′apai volcanic eruption and their global propagation in the mesosphere/lower thermosphere observed by meteor radars and modeled with the High-Altitude general Mechanistic Circulation Model, Atmos. Chem. Phys., 24, 4851–4873, https://doi.org/10.5194/acp-24-4851-2024, 2024.

Stober, G., Liu, A., Kozlovsky, A., Qiao, Z., Krochin, W., Shi, G., Kero, J., Tsutsumi, M., Gulbrandsen, N., Nozawa, S., Lester, M., Baumgarten, K., Belova, E., and Mitchell, N.: Identifying gravity waves launched by the Hunga Tonga–Hunga Ha′apai volcanic eruption in mesosphere/lower-thermosphere winds derived from CONDOR and the Nordic Meteor Radar Cluster, Ann. Geophys., 41, 197–208, https://doi.org/10.5194/angeo-41-197-2023, 2023.

Vadas, S. L., Becker, E., Figueiredo, C., Bossert, K., Harding, B. J., and Gasque, L. C.: Primary and secondary gravity waves and large-scale wind changes generated by the Tonga volcanic eruption on 15 January 2022: Modeling and comparison with ICON-MIGHTI winds, Journal of Geophysical Research: Space Physics, 128, https://doi. org/10.1029/2022JA031138, 2023.

Vadas, S. L., and Becker, E.: Numerical modeling of the excitation, propagation, and dissipation of primary and secondary gravity waves during wintertime at McMurdo Station in the Antarctic, Journal of Geophysical Research: Atmospheres, 123, 9326–9369. https://doi.org/10.1029/2017JD027974, 2018.

Vadas, S. L., Zhao, J., Chu, X., and Becker, E.: The excitation of secondary gravity waves from local body forces: Theory and observation, Journal of Geophysical Research: Atmospheres, 123, 9296–9325, https://doi.org/10.1029/ 2017 JD027970, 2018.

Becker, E., and Vadas, S. L.: Secondary gravity waves in the winter mesosphere: Results from a high-resolution global circulation model, Journal of Geophysical Reseach: Atmospheres, 123, 2605–2627, https://doi.org/ 10.1002/2017JD027460, 2018.